# Longevity gene responsible for robust blue organic materials employing thermally activated delayed fluorescence

Qing-Yu Meng[1,5], Rui Wang[1,5], Yi-Lei Wang[2], Xing-Wei Guo [2,3], Yu-Qi Liu[1], Xue-Liang Wen[1], Cheng-Yu Yao[1] & Juan Qiao [1,4] ✉

The 3rd-Gen OLED materials employing thermally-activated delayed fluorescence (TADF) combine advantages of first two for high-efficiency and low-cost devices. Though urgently needed, blue TADF emitters have not met stability requirement for applications. It is essential to elucidate the degradation mechanism and identify the tailored descriptor for material stability and device lifetime. Here, via in-material chemistry, we demonstrate chemical degradation of TADF materials involves critical role of bond cleavage at triplet state rather than singlet, and disclose the difference between bond dissociation energy of fragile bonds and first triplet state energy (BDE-$E_{T1}$) is linearly correlated with logarithm of reported device lifetime for various blue TADF emitters. This significant quantitative correlation strongly reveals the degradation mechanism of TADF materials have general characteristic in essence and BDE-$E_{T1}$ could be the shared "longevity gene". Our findings provide a critical molecular descriptor for high-throughput-virtual-screening and rational design to unlock the full potential of TADF materials and devices.

Organic light-emitting diodes (OLEDs) have become increasingly important to the cutting-edge flat-panel displays with higher quality and lower energy consumption. Particularly, the 3rd-Gen thermally activated delayed fluorescence (TADF)[1] materials have sparked considerable interest from academia and industry. In last decade, TADF materials have witnessed a flourishment in the conceptual advancements[2–10], and the external quantum efficiency (EQE) of TADF-OLEDs have recorded high up to 43.9%[10], being far beyond the early requirement of industrialization (>20%). However, operational lifetime of efficient blue TADF-OLEDs are still far from basic requirement[2,3,11], realistically a T95 (time to 95% of the initial luminance) of 500 h @1000 cd m$^{-2}$, impeding the commercialization of high-efficiency OLED devices[12] (Fig. 1a).

To date, it is generally considered that OLED devices degradation is mainly caused by chemical irreversible degradation of organic molecules at excited states[13–15] and for 2nd-Gen phosphorescent (PH)-OLEDs, the degradation is mainly induced by triplet-triplet annihilation (TTA) or triplet-polaron annihilation (TPA)[16]. As for 3rd-Gen TADF materials, considerable researchers take it granted that the degradation should also stem from TTA or TPA[2,3,11,17–23] like PH materials. Based on this, strategies improving operational lifetime such as reducing delayed lifetime ($\tau_d$)[3,11] and exciton energy[17,22], or decorating molecules with steric groups[17,18] have been proposed and proven effective in most cases. Meanwhile, some researchers hold the opposite view that the degradation process in singlet state could be more vital[24–26], since unlike PH materials, the final emission of TADF emitters is from singlet state not triplet. It seems that the degradation mechanism of TADF materials is still controversial and needs deeper insight.

Moreover, aforementioned strategies do not always work well. For example, though SpiroAC-Trz[27] has shorter $\tau_d$ (2.1 μs) and lower

[1]Key Lab of Organic Optoelectronics and Molecular Engineering of Ministry of Education, Department of Chemistry, Tsinghua University, Beijing, China. [2]Department of Chemistry, Tsinghua University, Beijing, China. [3]Center of Basic Molecular Science, Department of Chemistry, Tsinghua University, Beijing, China. [4]Laboratory for Flexible Electronics Technology, Tsinghua University, Beijing, China. [5]These authors contributed equally: Qing-Yu Meng, Rui Wang. ✉e-mail: qjuan@mail.tsinghua.edu.cn

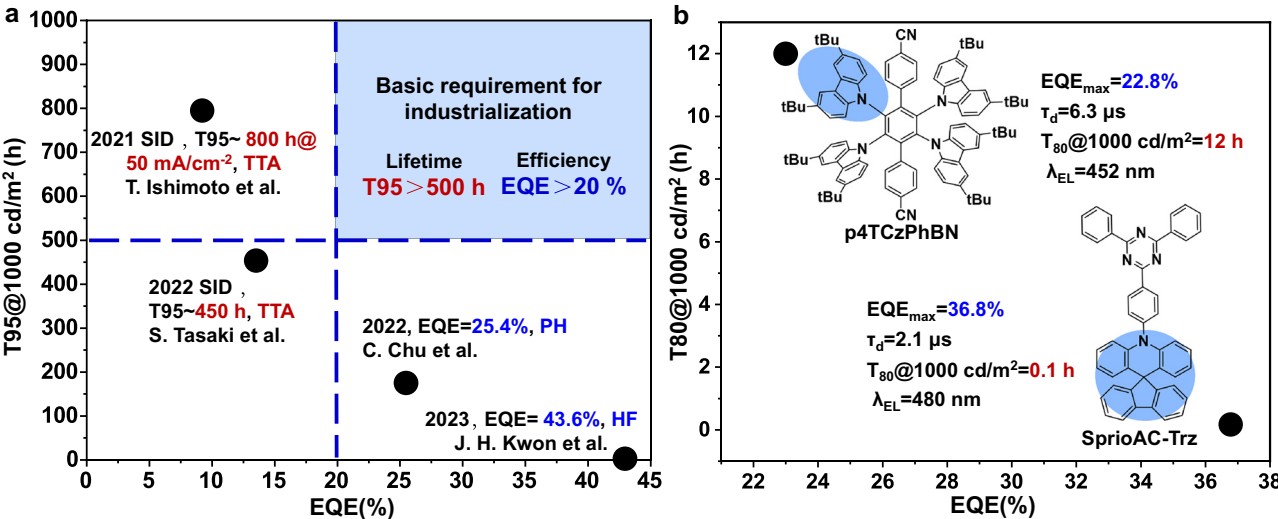

**Fig. 1 | Device performances of Blue OLEDs. a** Performances of TTA device based on low-efficiency, high-stability 1st-Gen fluorescence materials[50,51], PH device based on high-efficiency, low-stability 2nd-Gen blue PH materials[52] and hyperfluorescence (HF) device based on high-efficiency, low-stability 3rd-Gen TADF materials[10]. (Kwon et al. did not report the operational lifetime of HF device). **b** Chemical structures and device performances of two representative blue TADF emitters of p4TCzPhBN and SpiroAC-Trz.

exciton energy (480 nm, 2.58 eV) than p4TCzPhBN[2] (6.3 μs and 452 nm, 2.74 eV) (Fig. 1b), the optimized device lifetime of SpiroAC-Trz is only 1/120 of that of p4TCzPhBN (0.1 h vs. 12 h). There must be some other genetic characteristics of themselves, which might act as "Longevity Gene", affecting the intrinsic stability of blue TADF materials. Ever since Kondakov et al. revealed typical host material CBP suffered C-N bond dissociation in operational devices[28], bond dissociation energy (BDE) has been a key parameter for intrinsic stability of OLED materials[17,19,20,25,26,29–33], yet is often overlooked in molecule design. Furthermore, High throughput virtual screening (HTVS) is becoming a ground-breaking tool in screening TADF emitters with EQE values up to 22%[34]. However, Due to the elusive chemical degradation mechanism and lack of tailored descriptors of molecule stability, HTVS of robust blue TADF emitters have not implemented.

In this work, we systematically investigated the chemical degradation process of typical blue TADF emitters at excited states and demonstrated that bond cleavage of TADF materials at $T_1$ state is the main cause of their chemical degradation. Importantly, we revealed that the difference of BDE and triplet energy (BDE-$E_{T1}$) of TADF emitters is positively and linearly correlated with the logarithm of formation rate constant of quenchers causing by irreversible chemical degradation at $T_1$ state. Theoretical calculations on activation energy of the transition state during the bond cleavage solve the puzzle between thermodynamic parameter of BDE-$E_{T1}$ and kinetic degradation process. This unique quantitative correlation actually applies to the device lifetime of a wide variety of blue TADF emitters reported by different research groups in exactly different device structures. This finding strongly revealed the degradation mechanism at $T_1$ state have the general characteristic in essence and BDE-$E_{T1}$ has the ability to be the shared longevity gene responsible for robust TADF materials and devices. Our work would pave a new avenue for high-throughput screenings and rational design to attaining robust TADF emitters and speed up the iteration and commercialization of highly efficient and stable OLED material and devices.

## Results

### Degradation mechanism in excited states, singlet or triplet states?

Considerable works have demonstrated that the operational stability of TADF-OLEDs can be enhanced by improving the material stability at

singlet/triplet excited state[3,11,18–26,29–33,35–37]. However, it is yet undetermined that whether TADF emitters mainly degrade at singlet or triplet states, which still remains to be identified by rigorous and solid experiments based on a wide variety of TADF materials. Herein, we took six representative TADF materials, namely, DMAC-DPS[38], SpiroAC-Trz[27], DPAC-Trz[27], DCzTrz[37], DDCzTrz[37], and 5CzBN[18] (Fig. 2a) as research objects. These emitters include the most-widely used building blocks and all feature the most typical Donor-π-Acceptor (D-π-A) structures. They involve typical fragile bonds including C-S, and C-N single bond from different donors. BDE values of these fragile bonds (BDE$_f$[32]) vary from 3.57 eV to 4.43 eV with ~1 eV difference and the EL peaks vary from 459 nm to 500 nm, which guarantee the generality of conclusions.

First, we identified the degradation products of these emitters via Laser Desorption Ionization-Time of Flight Mass Spectroscopy (LDI-TOF MS)[30,39] and confirmed that they all suffer the cleavage of fragile bonds (Supplementary Fig. 1). Then, we conducted UV photo-degradation tests of DMAC-DPS, DPAC-Trz, and DCzTrz containing different fragile bond. We choose benzene as the right solvent and measured the emission spectra of these emitters (Supplementary Fig. 2). After 5 h in-situ continuous irradiation at 380 nm, emission intensity of these degassed solutions all decayed significantly, while those of aerated solutions nearly unchanged (Fig. 2b). This large disparity could be attributed to the different $S_1/T_1$ exciton dynamics in degassed and aerated solutions. In the degassed solution, $T_1$ excitons can exist and decay via non-radiation or thermally-activated reverse intersystem crossing (RISC) process from $T_1$ to $S_1$ excitons. Due to the two rate constants, $k_{nr,T}$ and $k_{RISC}$ of these emitters are comparable ($10^4$–$10^5$ s$^{-1}$, Supplementary Table 2), an equilibrium of conversion between $S_1$ and $T_1$ could be established. In contrast, in aerated solution, $T_1$ excitons are almost quenched by oxygen due to the large quenching rate constant (~$10^9$ s$^{-1}$)[40], which is 4–5 orders of magnitude larger than $k_{RISC}$ and $k_{nr,T}$. Thus, the equilibrium of conversion between $S_1$ and $T_1$ no longer exist and $T_1$ excitons almost vanish. Under the circumstances, we assumed these TADF materials may degrade mainly at triplet state. To confirm this assumption, we further conducted photo-degradation tests of degassed solutions with triplet quencher (2-Benzoylnaphthalene, 2-BP). 2-BP has higher $S_1$ energy (3.45 eV) and lower $T_1$ energy (2.47 eV) than these TADF emitters' (2.7–3.0 eV), so it cannot quench those $S_1$ excitons but efficiently quench $T_1$ excitons. Indeed,

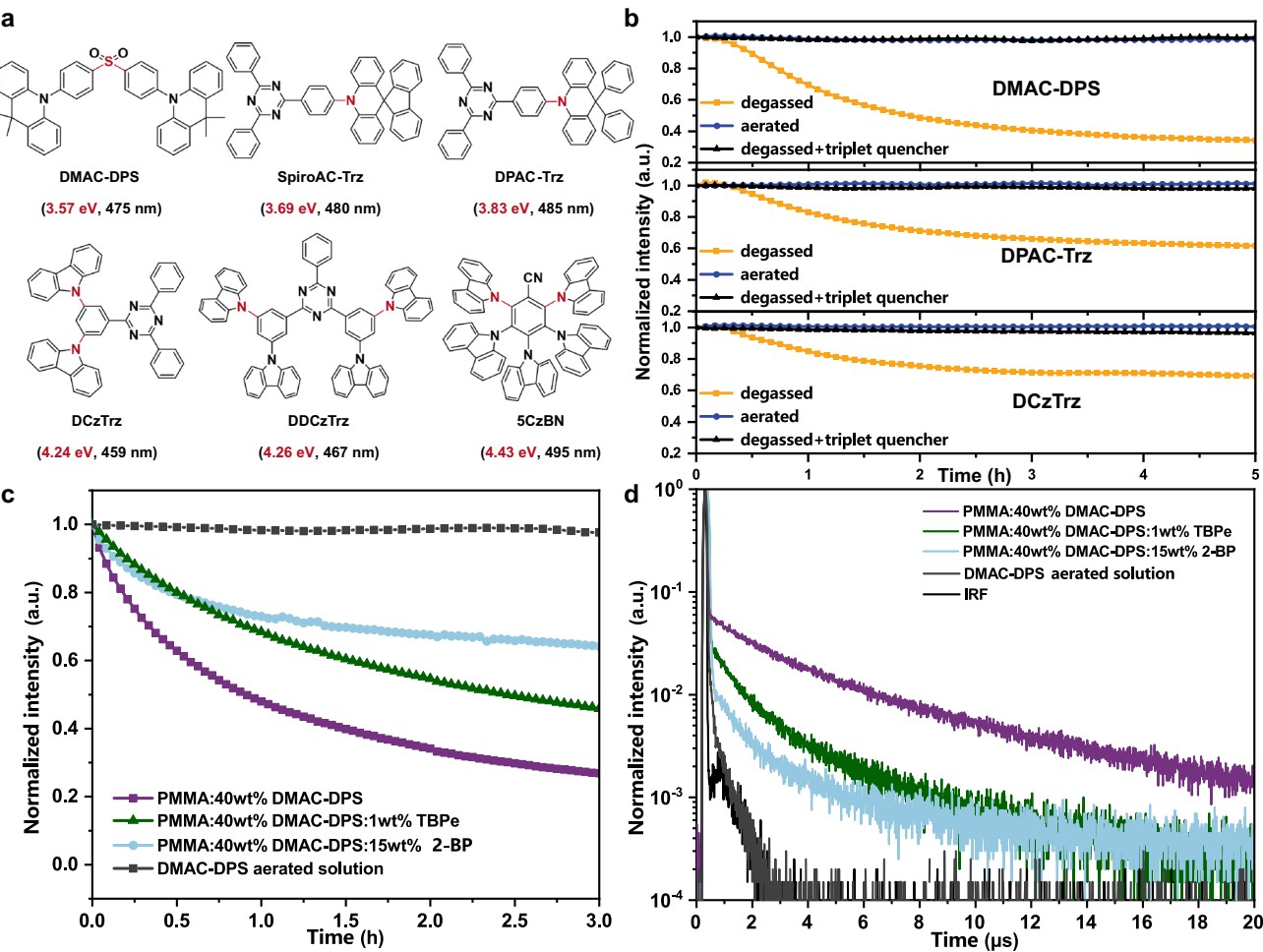

**Fig. 2 | UV photo-degradation test of TADF emitters. a** Chemical structures of typical blue TADF materials in this study, and the energy and wavelength in the parentheses corresponding to their BDE of fragile bonds and EL wavelengths values, respectively. The fragile bonds were labeled in red. **b** The 380 nm UV photo-degradation results of DMAC-DPS ($5.0 \times 10^{-5}$ M), DPAC-Trz ($3.9 \times 10^{-5}$ M), and DCzTrz ($7.1 \times 10^{-5}$ M) in benzene solutions (The concentration of each solution was determined by their absorbance with 0.107 at 380 nm). The concentration of the triplet quencher is $5.0 \times 10^{-4}$ M. **c, d** The 380 nm UV photo-degradation results (**c**) and transient PL spectra (**d**) of doped film (40 wt% DMAC-DPS: PMMA), TSF film (1 wt% TBPe: 40 wt% DMAC-DPS: PMMA), quencher-doped film (15 wt% 2-BP: 40 wt% DMAC-DPS: PMMA), and aerated solution of DMAC-DPS ($5.0 \times 10^{-5}$ M).

the emission intensity of each degassed solution with 2-BP were nearly unchanged (Fig. 2b), corroborating our assumption.

Next, to examine the effect of $T_1$ exciton in film state, we conducted photo-degradation measurements of DMAC-DPS in 40 wt% doped PMMA film, thermally-activated sensitized fluorescence (TSF) film with 1 wt% TBPe: 40 wt% DMAC-DPS: PMMA, and quencher-doped film with 15 wt% 2-BP: 40 wt% DMAC-DPS: PMMA. As shown in Fig. 2c, all the films decayed strikingly after 3 h UV photodegradation. To understand this result, we measured their transient PL spectra (Fig. 2d) and found all the films possess delayed emission from the conversion from $T_1$ to $S_1$. The order of their $\tau_d$ is doped film > TSF film > quencher-doped film, exactly opposing to their photostability, which confirmed TADF materials do degrade at $T_1$ state whatever in solution or films. Notably, the degradation in films cannot be completely suppressed even in TSF films or quencher-doped film because the emitter or quencher could not effectively diffuse in the film. More discussions were summarized in Supplementary Figs. 3–5.

**Numerical simulation of photodegradation tests of neat films**

After elucidating TADF materials mainly degrade at $T_1$ state, the question is how to quantitatively describe this degradation behavior. Freidzon et al.[19] once assumed the degradation rate constant ($k_D$) of host material in PH devices is related to BDE-$E_{exc}$ according to Arrhenius equation, where $E_{exc}$ referred to the energy depending on different degradation mechanism. However, this relationship has not been verified by experiments. Herein, we assumed "BDE – $E_{T1}$" would be a key molecule parameter describing the intrinsic stability of TADF materials at $T_1$ state. Then we performed numerical simulation to get the explicit and quantitative correlation. This method has been employed to study the exciton dynamics in TSF film[41] and TADF devices[21]. To exclude effects of other materials in devices, our simulation is based on neat films of TADF emitters by in-material chemistry rather than in-device chemistry. Based on the Jablonski diagram (Fig. 3), we derived the density of time-dependent excitons and exciton quenchers as Eqs. (1)–(3) (See detailed descriptions of exciton dynamics and kinetic Eqs. (1)–(3) in Supplementary Information).

$$\frac{dn_S}{dt} = I - (k_{r,S} + k_{nr,S} + k_{ISC})n_S + k_{RISC}n_T \\ - (k_{SSA}n_S^2 + k_{STA}n_S n_T - \gamma k_{TTA}n_T^2) - k_{QS}n_S n_Q \tag{1}$$

$$\frac{dn_T}{dt} = k_{ISC}n_S - (k_{r,T} + k_{nr,T} + k_{RISC})n_T \\ - (1+\gamma)k_{TTA}n_T^2 - k_{QT}n_T n_Q - k_{QF}n_T \tag{2}$$

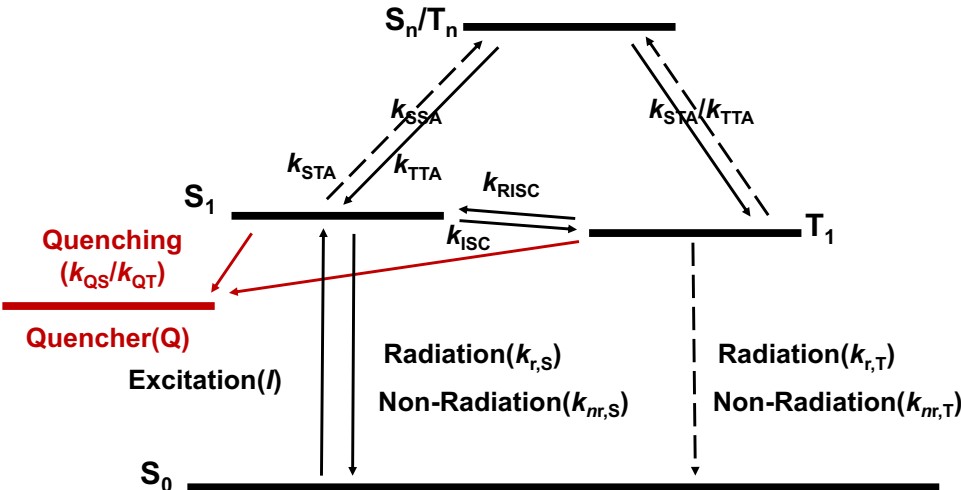

**Fig. 3 | Jablonski diagram of the exciton dynamics in TADF materials by photo-excitation.** $S_1$ excitons formed through photo-excitation would decay via radiative (prompt fluorescence) and non-radiative process, convert into $T_1$ excitons via intersystem crossing (ISC) process, or join singlet-singlet annihilation (SSA)/singlet-triplet annihilation (STA) process. $T_1$ excitons formed through ISC from $S_1$ would decay via radiative (phosphorescence) and non-radiative (degradation or thermal dissipation) process, convert into $S_1$ excitons via RISC process, or join STA/TTA process.

$$\frac{dn_Q}{dt} = k_{QF} n_T \qquad (3)$$

Here, $n_S$, $n_T$, and $n_Q$ represents the density of $S_1$, $T_1$ excitons and quenchers. $I$ is the intensity of excitation, which is set as a constant obtained from illumination intensity and transmittance. $k_{r,S}$, $k_{nr,S}$, and $k_{ISC}$ are rate constant of radiation, non-radiation, and intersystem crossing (ISC) of $S_1$ exciton. $k_{r,T}$, $k_{nr,T}$, and $k_{RISC}$ are rate constant of radiation, non-radiation, and RISC of $T_1$ exciton. $k_{SSA}$, $k_{STA}$, and $k_{TTA}$ are rate constant of singlet-singlet annihilation (SSA), singlet-triplet annihilation (STA), and TTA process. $\gamma$ is the spin factor of 0.25[42]. $k_{QS}$ and $k_{QT}$ are quenching rate constant of $S_1$ and $T_1$ exciton by quenchers. $k_{QF}$ is the formation rate constant of quencher caused by irreversible photo-induced deterioration, manifesting the intrinsic material stability. All parameter choices and their rationality are summarized in Supplementary Fig. 6 and Supplementary Table 2. Based on the given parameters, we got high-quality fitting curves of each neat film (Fig. 4a and Supplementary Fig. 7a), and derived $k_{QS}$, $k_{QT}$ and $k_{QF}$ values of each emitter (Supplementary Table 2). Since the calculated photophysical parameters depend on whether non-radiative decy from $S_1$ or $T_1$ is assumed to be 0[43], we got the two $k_{QF}$ values of each material with the assumption of $k_{nr,T} = 0$ and $k_{nr,S} = 0$, respectively.

Firstly, according to simulation results, we excluded the effects of SSA, STA and TTA processes on the quencher formation in our study. We simulated the dependence of each photophysical process rate on the intensity of illumination. At the test condition of illumination intensity ~2 mW cm$^{-2}$ (Fig. 4b), the rate of SSA, STA and TTA is 4−7 order of magnitude slower than that of radiation, RISC, etc., which indicates the quenchers formed through SSA, STA, and TTA are negligible. Take TTA as an example, we conducted the simulation with/without TTA process under the assumption of $k_{nr,T} = 0$ and assumption of $k_{nr,S} = 0$, respectively (Supplementary Fig. 8). Regardless of $k_{nr,T} = 0$ or $k_{nr,S} = 0$, the simulation results with/without TTA process are nearly unchanged, which indicates TTA process have little effect on the degradation of materials in our work. Furthermore, we took DMAC-DPS as an example and compared the simulation curve based on the assumption quenchers formed from single excitons (single exciton model) with those based on the assumptions that quenchers originating from hot excitons induced by SSA, STA, or TTA (hot-exciton model). As shown in Fig. 4c, no matter how we adjusted the parameters in hot exciton model, the model could only fit well either at initial part (hot exciton model 1) or ending part (hot exciton model 2), which further supports that hot exciton process is not the main quencher formation way in our study. (Detailed discussions are in Detailed discussions for the comparison between single exciton model and hot exciton models in Supplementary Information).

Secondly, we explored the relationship between $k_{QF}$ and BDE-$E_{T1}$ and found the logarithm of $k_{QF}$ is negatively and linearly correlated with the value of BDE-$E_{T1}$ whether $k_{nr,T}$ (Supplementary Fig. 7c) or $k_{nr,S}$ (Supplementary Fig. 7d) is assumed to be 0, and the logarithm of the average value of $k_{QF}$ is correlated with BDE-$E_{T1}$ with $R^2 = 0.83$ (Fig. 4d). These results clearly demonstrate that BDE-$E_{T1}$ is really one key parameter determining the molecular stability of blue TADF materials at triplet states.

## Theoretical calculations on dynamic process of bond cleavage at $T_1$ state

Considering again the correlation between BDE-$E_{T1}$ and $k_{QF}$, a fascinating question naturally arises that how come the thermodynamic parameter BDE-$E_{T1}$ strongly correlated with the kinetic parameter, $k_{QF}$. There must be some underlying correlations between BDE-$E_{T1}$ and the bond cleavage process of TADF materials. Recently, Adachi et al.[44] and Ihn et al.[35] both found transition states during the bond cleavage process of some TADF molecules at $T_1$ state. According to Bell-Evans-Polanyi principle[45], we speculated the existence of transition state may link the above thermodynamic parameter and dynamic process. Thus, we investigated the corresponding C-X bond (X = S, N) cleavage at $T_1$ state via quantum-chemical calculations. To allow for tractable computations, we ignored the specific structure of acceptors and constructed corresponding D-π-A model molecules (Fig. 5a), 4,4'-sulfonyldianiline (ADPS), 4-(10H-spiro[acridine-9,9'-fluoren]–10-yl) benzonitrile (Spiro-AC-CN), 4-(9,9-diphenylacridin-10(9H)-yl) benzonitrile (DPAC-CN), and 4-(9H-carbazol-9-yl) benzonitrile (CzCN). Optimized structures are summarized in Supplementary Fig. 12 and Supplementary Table 3. The four molecules show similar bond cleavage process that the energy first rises as the bond elongation and reaches maximum at 0.4−0.5 Å and then decreases monotonically, clearly manifesting the existence of transition state (Fig. 5b).

Then, we optimized structures at the energy maximum point (Supplementary Fig. 13) and got the frontier molecule orbitals of

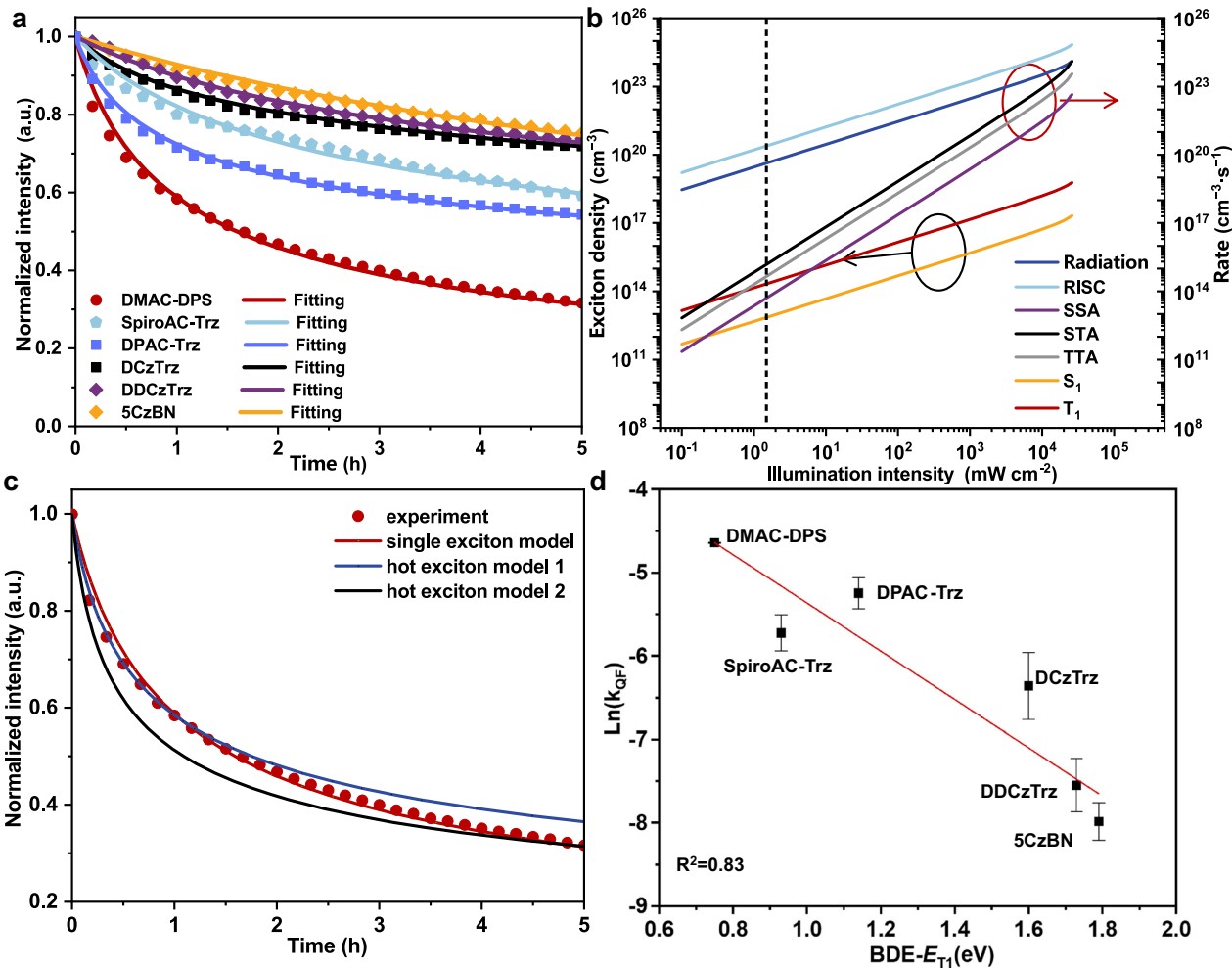

**Fig. 4 | Numerical simulation of photodegradation curves of neat films.**
**a** Experimental and simulation results on neat films of DMAC-DPS, SpiroAC-Trz, DPAC-Trz, DCzTrz, DDCzTrz, and 5CzBN with the thickness of 80 nm. **b** The influence of illumination intensity on exciton density and photophysical parameters just before quencher formation (The experimental illumination intensity ~2 mW cm⁻² is labeled by the black dot line). **c** Comparison of different degradation

models. All the results were simulated with the assumption of $k_{nr,S} = 0$. **d** The correlation between the average quencher formation rate ($k_{QF}$) and BDE-$E_{T1}$. The error bar values refer to the two different $k_{QF}$ values with the assumption of $k_{nr,T} = 0$ (the upper bar) or $k_{nr,S} = 0$ (the below bar). Source data are provided as a Source Data file.

transition state (Fig. 5c). Transition state of ADPS was previously reported to feature coupling of $^3\pi\text{-}\pi^*$ state and $^3\sigma\text{-}\sigma^*$ state[44]. Indeed, the Lowest Single Occupied Orbital (LSOMO) and Highest Single Occupied Orbital (HSOMO) of $T_1$ state are mainly $^3\pi\text{-}\pi^*$ state while those of transition state show admixture of $^3\pi\text{-}\pi^*$ (located on phenyl) and $^3\sigma\text{-}\sigma^*$ (located on S-C[1] bond) characteristics. Similar coupling also occurs in SpiroAC-CN, DPAC-CN, and CzCN. Thus, according to Bell-Evans-Polanyi relation[45], the activation energy ($E_a$) of C-S/C-N single bond cleavage process should be positively and linearly correlated with the energy difference between the final and initial state, that is just BDE-$E_{T1}$ (Fig. 5d). To examine this correlation, we calculated BDE-$E_{T1}$ and $E_a$ of corresponding bond cleavage process for a wide variety of model molecules (Supplementary Fig. 14). Remarkably, they indeed show a perfect linearity correlation with $R^2 = 0.98$ (Fig. 5e). This result exactly bridges the gap between thermodynamic parameter of BDE-$E_{T1}$ and kinetic process of quencher formation.

**Effect of "BDE − $E_{T1}$" on the operational lifetime of OLED devices**
It is well-accepted that material stability has crucial influence on device lifetime, and researchers are most committed to exploring a general descriptor of material stability to describe or even predict the operational lifetime of corresponding OLED devices. However, to date, such

desired descriptor has never been sought out. So, to approach this great desire, we tried to correlate BDE-$E_{T1}$ with device lifetime of DMAC-DPS, SpiroAC-Trz, DPAC-Trz, DCzTrz, DDCzTrz, and 5CzBN reported in literatures. Encouragingly, the logarithm of device lifetime is likewise positively and linearly correlated with BDE-$E_{T1}$ and $R^2$ is as high as 0.92 (Fig. 6a), which strongly suggested BDE-$E_{T1}$ should be a key thermodynamic parameter determining device lifetime except for kinetic parameters previously reported.

To confirm this correlation, we further collected TADF materials (35 points) reported with operational lifetime and device emission peak <500 nm (Supplementary Fig. 15 and Supplementary Table 4). Unexpectedly, the correlation of them did not reach a significant level (Supplementary Fig. 16) with $R^2 = 0.44$. However, for all the materials, the general trend remains the higher BDE-$E_{T1}$ is, the longer device lifetime is. Particularly, we carefully examined the materials exhibited exceptionally longer lifetime beyond the correlation. For DBA-DI and TDBA-DI reported by Kwon et al.[20,46], although their device lifetime is higher than predicted, DBA-DI with higher BDE-$E_{T1}$ indeed showed longer device lifetime. Similar phenomenon occurred in TMCzTrz, 5CzTrz, and DACT-II reported by Adachi et al.[3] Also, DBA-DI, TDBA-DI, and 5CzTrz, all possess high $k_{RISC}$ values ($6.21 \times 10^6$ s⁻¹, $1.08 \times 10^6$ s⁻¹, and $1.5 \times 10^7$ s⁻¹, respectively). Therefore, the longer operational

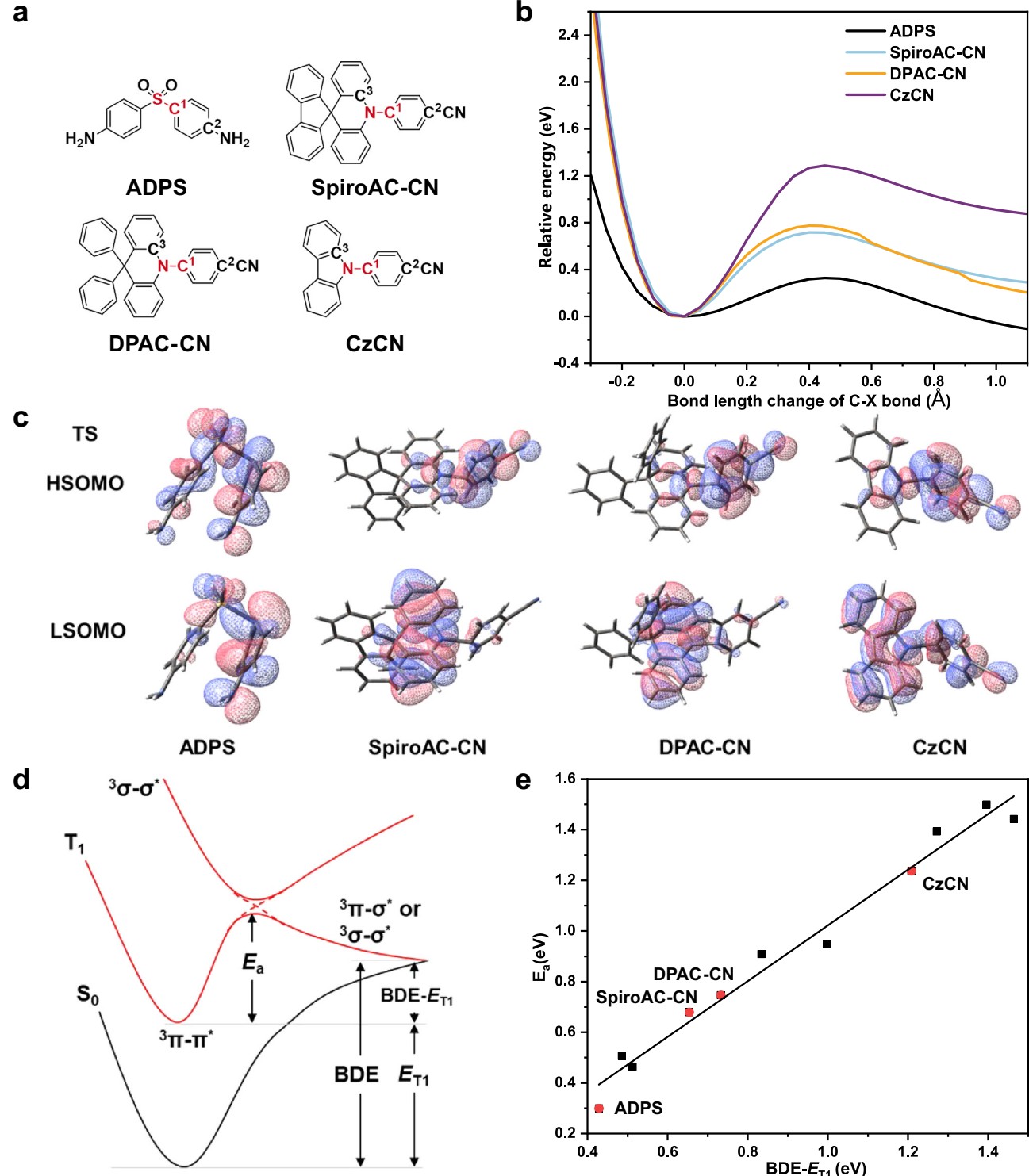

**Fig. 5 | Dynamic study of bond cleavage at T1 state. a** Chemical structures of model molecules, namely, ADPS, SpiroAC-Trz, DPAC-CN, CzCN. **b** The potential energy curves (PEC) of triplet excitons of model molecules along the C-X bond dissociation coordinate. **c** Chemical structures, and frontier molecule orbitals of transition state of model molecules. **d** PEC of C-X (X = N, S et al.) bond cleavage at T1 state. For most of TADF materials, T1 state is $^3\pi$-$\pi^*$ state, and the state corresponding to C-X single bond cleavage is $^3\sigma$-$\sigma^*$ state. When a molecule is excited to T1 state from S0 state and the C-X bond is elongated, the $^3\pi$-$\pi^*$ state and $^3\sigma$-$\sigma^*$ state would be coupled at the crossing point corresponding to the transition state (TS). The energy required to reach TS from the optimized structure of T1 state is $E_a$. **e** The correlation between BDE-$E_{T1}$ and $E_a$ in a wide variety of TADF model molecules.

lifetime of these 3 materials could also be partially attributed to the superior $k_{RISC}$ and the prudent device optimization and engineering. On the other hand, we found 9 materials reported by Zhang et al.[24,25] also did not fit the correlation well. These materials have very comparable BDE values (~4.2 eV for carbazole derivatives and ~3.6 eV for

DMAC derivatives) and $E_{T1}$ values (~2.7 eV), which leads to very comparable BDE-$E_{T1}$ values (~1.5 eV for carbazole derivatives and ~0.9 eV for DMAC derivatives). Therefore, in these materials, the difference in device lifetime is no longer decided by BDE-$E_{T1}$, but other factors. Indeed, Zhang et al. explained the different device lifetime by different

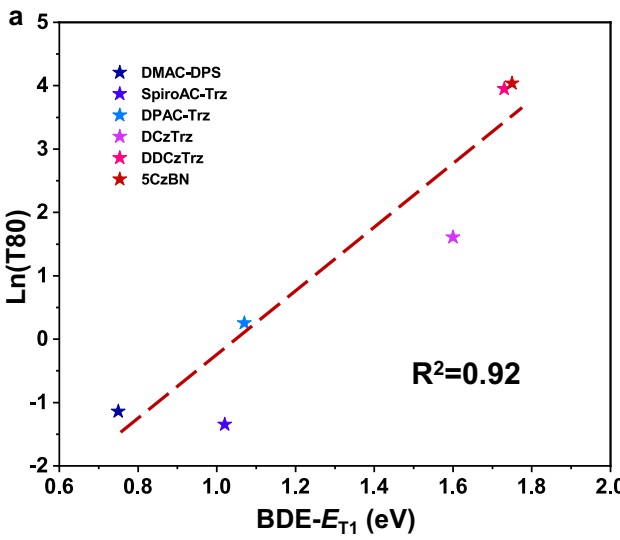

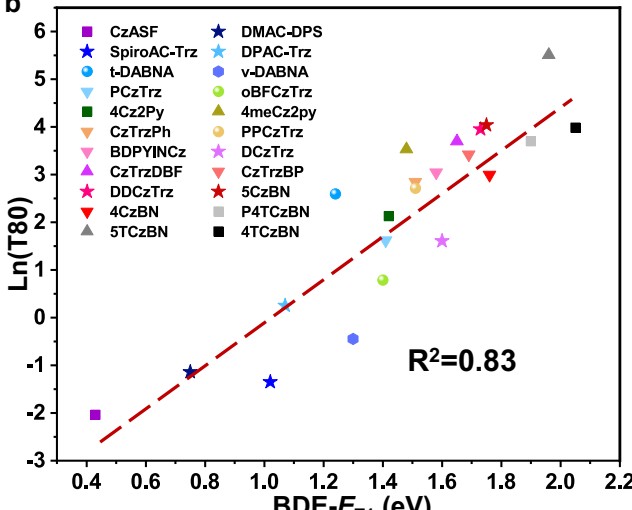

**Fig. 6 | The correlation between BDE -$E_{T1}$ and device lifetime of TADF materials reported in literatures. a** Correlation in DMAC-DPS, SpiroAC-Trz, DPAC-Trz, DCzTrz, DDCzTrz, 5CzBN studied in the photo-degradation section. **b** Correlation in more TADF emitters reported with operational lifetime.

$k_f$ and EL emission wavelength of those carbazole derivatives in different host materials[24] (See Discussions of materials exhibited exceptionally longer lifetime beyond the correlation in supplementary information for more detailed discussions.). Thus, to get more general conclusion, we excluded these materials and surprisingly found $R^2$ returned to 0.83 for remaining 22 points (Fig. 6b). Mind that these remaining TADF materials have very different molecular structures and were reported by different research groups, this significant quantitative correlation strongly revealed the degradation mechanism of TADF materials have general characteristic in essence and BDE-$E_{T1}$ has the capability to be the shared longevity gene responsible for robust TADF materials and devices. To deepen the insight into this correlation, we sought to make further analysis and discussions as follows.

## Discussion

Firstly, we emphasized that BDE-$E_{T1}$ is not the one and only molecule parameter affecting TADF material stability and device lifetime, but can act as the intrinsic one. Recently, Lee et al. demonstrated a good linearity correlation between $k_{RISC}$ of TADF emitters and corresponding device operational lifetime[33]. Yet, this correlation was obtained from 4 TADF emitters featuring very similar molecule structures. To explore its generality, we tried to collect $k_{RISC}$ values of the above 35 points from literatures, but got only that of 13 points because the others were not reported. Unfortunately, we didn't find equally good linearity correlation in such a wide variety of materials (Supplementary Fig. 18a). It might originate from the high sensitivity of kinetic parameter $k_{RISC}$ value to different measurement and calculation methods[43]. In comparison, the thermodynamic parameter BDE-$E_{T1}$ is relatively insensitive and thus could act as the "longevity gene" for the "trait" of device lifetime. Certainly, during "gene expression", "environment" such as carrier balance and host-guest interactions, would also have important influence on device lifetime, because they could greatly affect the kinetic parameters. Thus, the surprisingly linear correlation in Fig. 6b might also thank to the host are almost mCBP or its derivatives and the reported device structures have been optimized aiming to the balance carrier for high EQE.

Secondly, although the BDE values are calculated in neutral state, some work reported that BDE in anion state is more important for device lifetime[20,21]. In fact, BDE in anion state is highly related to BDE

in neutral state according to Hess's law[47]. In addition, Kaji et al. reported that since EQEs of reported devices are almost >20%, the charge carrier balance could be considered to be well settled in devices and the effects of cationic/anionic states are considered small[29]. Moreover, for blue TADF emitters with $E_{T1}$ between 2.7–2.9 eV, their BDE-$E_{T1}$ is highly linearly correlated to BDE-2$E_{T1}$ with $R^2 = 0.96$ (Supplementary Fig. 17). Thus BDE-$E_{T1}$ could also roughly reflect the molecule stability in TTA process. Therefore, in principle, the "longevity gene", BDE-$E_{T1}$ could not only describe the intrinsic material stability in excited state, but also characterize and even predict the operational lifetime of optimized devices, manifesting its ability being a desired molecular descriptor in HTVS and material design efforts.

Thirdly, answering the stability puzzle between SpiroAC-Trz and p4TCzPhBN in the introduction, it is much better "longevity gene", BDE-$E_{T1}$ that leads to much longer device lifetime of p4TCzPhBN. Similar cases exist in other TADF materials based on carbazole and acridine (Fig. 6b), which further confirms the intrinsic effect of "longevity gene" on the material stability and corresponding device lifetime. To enhance the "longevity gene" BDE-$E_{T1}$, one could start with two aspects, i.e., lowing $E_{T1}$ or improving BDE. For example, Adachi et al. introduced pyrene unit into MR-TADF material BCzBN, significantly lowering its $E_{T1}$ and improving the operational stability of corresponding material and device[22]. Considering the relatively fixed $E_{T1}$ energy for highly efficient blue emission, we recommend that BDE, which is often overlooked before, deserves more attention to. In the last decade, we focused on the BDE and intrinsic stability of blue organic emitters and drew some valuable design strategies for robust blue materials[30,32,47]. We believe that effective strategy to improve BDE without negative effects on material photophysical properties would be highly desired in the future development of robust blue TADF materials.

In summary, for the 3rd-Gen OLED materials, it is still an unsettled issue that blue TADF materials have not met the basic stability requirement for practical applications, which is essentially due to the ambiguous degradation mechanism and the lack of appropriate and quantitative parameters describing material stability. To address this issue, via in-material chemistry, we figure out the underlying chemical degradation mechanism that TADF materials mainly degrade at $T_1$ state rather than $S_1$. Most importantly, we uncover that BDE-$E_{T1}$ as the critical molecule parameter of TADF

emitters is positively and linearly correlated with the logarithm of quencher formation rate $k_{QF}$ and this correlation also applies to corresponding device lifetime for a wide variety of TADF emitters. To the best of our knowledge, this study for the first time identifies the "longevity gene" for robust TADF materials, which could be particularly valuable for HTVS in material development. Our findings would unlock the full potential of organic emitters employing TADF by inserting the "longevity gene", that is high BDE-$E_{T1}$, to speed up the iteration and commercialization of highly efficient and stable blue OLED materials and devices for more tremendous applications in high-end displays and lighting.

## Methods

### Materials
The reported SpiroAC-Trz, DCzTrz, DDCzTrz, 5CzBN were synthesized and characterized according to the corresponding literature[18,27,37]. 2-BP was purchased from J&K Scientific Co., Ltd. TBPe was provided by Eternal Material Technology Co., Ltd. DPAC-Trz, and TPPDA was provided by Prof. Qi-sheng Zhang from Zhejiang University.

### General information
$^1$H NMR spectra were measured on JNM-ECZ600R 600-MHz NMR spectrometer using $CDCl_3$ as the solvent and tetramethyl silane as an internal standard at room temperature. LDI-TOF-MS measurements were conducted on Shimadzu AXIMA Performance. The applied voltage between the target and the TOF aperture is 25 kV. The sample powder was dissolved by dichloromethane without an assistant matrix. After solvent evaporation, the samples were excited by the pulsed nitrogen laser beam (337 nm) with a spot size of 0.01 mm$^2$.

### Steady-state spectra, lifetime and photoluminescence quantum yield measurements
Steady-state absorption and emission spectra were recorded by Lengguang Technology UV-1920Pro and Hitachi F-7000, respectively. The photoluminescence quantum yields were obtained by absolute method on Hamamatsu C9920. The time-resolved electroluminescence measurements were conducted on Edinburgh FL920P.

### Sample preparation and measurement procedures of UV degradation
For UV degradation measurements, the indium tin oxide (ITO)-coated glass substrates were precleared and treated by UV-ozone for 30 min. The evaporation processes were performed at a pressure under $1 \times 10^{-4}$ Pa. The deposition rate for organic materials is 0.1 nm/s. The UV degradation measurement were carried by Hitachi F-7000 Fluorescence Spectrometer.

### Quantum-calculation and numerical simulation
Calculations and analysis in this work were performed with Gaussian 16[48], and Multiwfn (3.7) [49]. BDE was calculated as the energy changes of the bond cleavage reactions at 298.15 K and 1 atm (gas phase). The corresponding geometry optimizations and frequency analysis were performed at the density functional theory (DFT) level using M06-2X functional with dispersion correction (D3) and 6-31+G* basis set. The scan of potential energy surface for bond cleavage process and the optimized of transition state were performed at wB97X-D/6-31G* level. Numerical simulations were conducted by Matlab. More details were summarized in the supplementary discussions for numerical simulation section in supplementary information.

## Data availability
The authors declare that the data supporting the findings of this study are available within the paper and its supplementary information file. Source data are provided with this paper.

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

## Acknowledgements

This work was financially supported by the National Key R&D Program of China (No. 2022YFB3603002) and the National Natural Science Foundation of China (No. 52273184). We are grateful to Prof. Qi-Sheng Zhang from Zhejiang University for his provision of DPAC-Trz and TPPDA. We thank the Tsinghua National Laboratory for Information Science and Technology and Tsinghua Xuetang Talents Program for supporting the computational resources.

## Author contributions

J.Q. conceived the project. R.W. and Q.M. synthesized organic compounds used in this study and prepared the samples and measured their properties. Q.M., R.W, and Y.L. performed the UV degradation tests. R.W. carried out the quantum-chemical calculations and numerical simulations. Q.M. and C.Y. collected and analyzed the device lifetime data in literatures. Q.M. wrote the paper. X.G. and Y.W. provided inspiring suggestions for improvement. X.W. provided helpful discussions. J.Q. and Y.W. supervised the project. All authors commented on the paper.

## Competing interests

The authors declare no competing interests.
