## [Peer Review File · Nature Communications]

Longevity Gene Responsible for Robust Blue Organic Materials Employing Thermally Activated Delayed FluorescenceREVIEWER COMMENTS

Reviewer #1 (Remarks to the Author):

Juan Qiao et al proposed a way of degradation prediction in blue TADF materials for device stability improvement. Prior to this, they conducted an experiment to compare PL intensity reduction caused by material degradation to identify the main cause of the degradation mechanism. After 5hrs in-situ continuous irradiation at 380nm, emission intensity of these degassed solutions all decayed significantly, while those of aerated solutions nearly unchanged. Next, to examine the effect of T1 exciton in film state, they conducted photo-degradation measurements of TADF film, TSF film, triplet quencher-doped film. The order of their photostability is TADF film > TSF film > triplet quencher-doped film, which confirmed TADF materials do degrade at T1 state whatever in solution or film. They calculated the parameter k_{QF} using numerical simulation. Then they explored the relationship between k_{QF} and BDE-E_T1 and found the logarithm of k_{QF} is negatively and linearly correlated with the value of BDE-E_T1 with $R^2=0.85$. In addition, this new parameter "BDE-E_T1" show a good linearity correlation with $R^2=0.98$. This result exactly bridges the gap between thermodynamic parameter of BDE-E_T1 and kinetic process of quencher formation. Finally, they uncover that BDE-E_T1 as the critical molecule parameter of TADF emitters is positively and linearly correlated with the logarithm of device lifetime for a wide variety of TADF emitters.

BDE-E_T1 is a new parameter in TADF materials. However, many researchers already knew about high BDE and low E_T1 is important for long lifetime. So, if authors want to highlight the manuscript better to find the solution of how to enhance the high BDE-E_T1 in the T1 states that's more important for our current research (in OLEDs). As a reviewer feels that in this work does not contain any interesting results Therefore, I do not recommend this manuscript to be published in this journal.

1. Authors mentioned the numerical simulation excluded the rate of SSA, STA, TTA because of lower rate compared to radiation, rISC, etc. However, there is no sufficient explanation of what changes occur in the simulation results, when the hot excitons are formed by TTA, TSA, SSA.
2. When authors check the correlation between BDE-E_T1 and device lifetime of 32 reported materials, the value of R^2 is just 0.44. Authors said it is because of device optimization and engineering or the specific molecule structures. Please explain why these materials have different results.
3. Authors, please check the supporting page S21 line 279; I think Fig. 3c should be changed to Fig 4c.

Reviewer #2 (Remarks to the Author):

Stable and long lifetime triplet-harvesting blue OLEDs and OLED emitters (phosphorescent, TADF etc.) have been a long sought but un-solved issue for both research and industry. This work conduct fundamental and comprehensive studies on possible fundamental mechanisms of material factors potentially governing operation lifetimes of triplet-harvesting blue OLEDs and OLED emitters, particularly the advanced TADF blue emitters, and report an interesting discovery on the correlation between the difference between bond dissociation energy of fragile bonds and the first triplet state energy (BDE-ET1) and blue TADF emitter/OLED operation lifetimes. This reviewer finds this is a significant finding and contribution that would be highly useful for guiding development of highly stable blue TADF emitters and OLEDs. The results and evidence provided are considered convincing. I would recommend publication of this contribution.

Reviewer #3 (Remarks to the Author):

The manuscript provided the correlation between theoretical calculated bond dissociation energy, exciton lifetime, photo-stability and OLED device stability. To improve the stability especially on the blue OLED, this type study attracts much interest from OLED researchers. However, there are several questions as below. To publish in the Nature communications, the major correction would be necessary.

1. Basically, the delayed emission quenching of TADF is not perfect in aerated condition. It is because that the triplet quenching is basically electron exchange between excited molecule and oxygen as explained in Dexter energy transfer, therefore, the concentration should be an important factor. The authors explained as "2-BP has higher S1 energy (3.45 eV) and lower T1 energy (2.47 eV) than these TADF emitters' (2.7~3.0 eV), so it cannot quench those S1 excitons but efficiently quench T1 excitons. Indeed, the emission intensity of each degassed solution with 2-BP were nearly unchanged (Fig 2b), corroborating our assumption" in main text. This explanation may induce misunderstanding it related to be the energy state. The quencher concentration for solution state for Fig 2b should be provided. Also, the quencher should not diffuse in the film, which makes the difference between in solution and film states. Therefore, these result in solution and films should not be compared. In addition, the quencher is 2-BP in the main text and figure caption but NPK in figure.

2. About Fig S3 and S4, authors compared the diluted ($\lambda_{380} = 0.167$) and concentrated ($\lambda_{380} = 0.993$) to check the influence of singlet exciton for photo-degradation. It is well known things that the photo degradation occurs from triplet exciton mainly because of its longer lifetime than singlet exciton. The reaction probability is related not only the concentration but also the excitation state lifetime, therefore, it should be measured with the higher S1 exciton density than that of T1 (during all

the time for steady state excitation) with the consideration of those time dependency. While the resulted conclusion might not be different, is the measurement system satisfied this condition? In addition, reviewer have worried about its measurement system. When absorbance around 1 with 10×10 mm² cuvette was employed, it provides the large gradation of exciton concentration in solution. This would affect to the photo-degradation result. The measurement should be investigated with the same absorbance with the difference light path length. For example, it is necessary to use the cell having the light path length as 10 mm for diluted condition and 1 mm for concentrated condition in the case of 10 times difference concentration.

3. The ISC and RISC rate constants are affected by how PF/DF efficiencies are estimated, and whether non-radiative decay from S1 or T1 is assumed to be 0. Authors assuming the singlet non-radiative decay as 0 but it is reasonable to estimate with triplet non-radiative decay as 0 when radiative decay rate was assumed as 0. For more detailed discussion, it would be better to discuss with the range of rate constants. The detail of photophysical analysis for TADF was reported in J. Phys. Chem. A 2021, 125, 8074-8089. Especially, the materials showing 100% internal quantum efficiency in OLED have no non-radiative decay from triplet as reported in Front. Chem. 2022, 10, 990918.

4. About the hot-exciton model: The Fig 3 was illustrated as all of SSA, STA, TTA provide the S1 excitons however only TTA provided the S1 exciton in equation 1. Also, it would not suitable TTA efficiency was assumed as 0.25 because the TT up-conversion efficiency related to be each molecule. How does the simulation result obtain when TT up-conversion does not happen? Also, there are no term for $k(QF)n(T)$ in equation 2.

5. The experimental photostability was investigated in neat film but the material was doped in host-matrix in the devise. Also, the calculation is the mono molecular state. Is it possible to compare them directly?

6. There is the mistake in the reference number. Also, the style of references is not unified. It should be check again carefully.

Point-by-point Response to the Reviewers' comments

Manuscript ID: NCOMMS-23-14158-T

We sincerely thank the Referees for their insightful comments, constructive suggestions, and the time spent in reviewing our manuscript. Please find below our point-by-point responses to the comments. To ease readability, we are repeating the referee's comments in *italics*.

Reviewers' Comments:

Reviewer #1 (Remarks to the Author):

Juan Qiao et al proposed a way of degradation prediction in blue TADF materials for device stability improvement. Prior to this, they conducted an experiment to compare PL intensity reduction caused by material degradation to identify the main cause of the degradation mechanism. After 5hrs in-situ continuous irradiation at 380nm, emission intensity of these degassed solutions all decayed significantly, while those of aerated solutions nearly unchanged. Next, to examine the effect of T_1 exciton in film state, they conducted photo-degradation measurements of TADF film, TSF film, triplet quencher-doped film. The order of their photostability is TADF film > TSF film > triplet quencher-doped film, which confirmed TADF materials do degrade at T_1 state whatever in solution or film. They calculated the parameter k_{QF} using numerical simulation. Then they explored the relationship between k_{QF} and $BDE-E_{T1}$ and found the logarithm of k_{QF} is negatively and linearly correlated with the value of $BDE-E_{T1}$ with $R^2=0.85$. In addition, this new parameter “ $BDE-E_{T1}$ ” show a good linearity correlation with $R^2=0.98$. This result exactly bridges the gap between thermodynamic parameter of $BDE-E_{T1}$ and kinetic process of quencher formation. Finally, they uncover that $BDE-E_{T1}$ as the critical molecule parameter of TADF emitters is positively and linearly correlated with the logarithm of device lifetime for a wide variety of TADF emitters.

Responses:

We are grateful for the Reviewer highlighting the key findings of our work.

As mentioned by the Reviewer, we proposed a way of degradation prediction in blue TADF materials for device stability improvement and a new parameter “ $BDE-E_{T1}$ ”, which is as the critical molecule parameter of TADF emitters positively and linearly correlated with the logarithm of device lifetime for a wide variety of TADF emitters. As Reviewer #2 commented, “*This work conducts fundamental and comprehensive studies on possible fundamental mechanisms of material factors potentially governing operation lifetimes of triplet-harvesting blue OLEDs and OLED emitters,*

particularly the advanced TADF blue emitters, and report an interesting discovery on the correlation between the difference between bond dissociation energy of fragile bonds and the first triplet state energy ($BDE-E_{T1}$) and blue TADF emitter/OLED operation lifetimes. This is a significant finding and contribution that would be highly useful for guiding development of highly stable blue TADF emitters and OLEDs.”. And Reviewer #3 pointed out that “to improve the stability especially on the blue OLED, our study attracts much interest from OLED researchers”. Therefore, we believe our work should provide new insight into the degradation of blue TADF materials and accelerate the development of robust blue OLEDs materials.

BDE- E_{T1} is a new parameter in TADF materials. However, many researchers already knew about high BDE and low E_{T1} is important for long lifetime. So, if authors want to highlight the manuscript better to find the solution of how to enhance the high BDE- E_{T1} in the T_1 states that’s more important for our current research (in OLEDs). As a reviewer feels that in this work does not contain any interesting results Therefore, I do not recommend this manuscript to be published in this journal.

We totally agreed with the Reviewer’s statement that *BDE- E_{T1} is a new parameter in TADF materials*. As Reviewer #2 commented, “*This work... report an interesting discovery on the correlation between the difference between bond dissociation energy of fragile bonds and the first triplet state energy ($BDE-E_{T1}$) and blue TADF emitter/OLED operation lifetimes*”. But we partially agree on the reviewer’s statement that *many researchers already knew about high BDE and low E_{T1} is important for long lifetime*. Certainly, the chemical degradation in triplet states in TADF materials has been attached much importance to, but it is more like an analogy from the degradation of phosphorescence materials. The main cause of TADF materials degradation is still undetermined, remaining to be identified by rigorous and solid experiments. And compared with the importance paid to T_1 state, fewer researchers pay attention to the essential role of BDE in the degradation of TADF materials. Nowadays, to obtain high efficiency TADF materials, many researchers usually employ building blocks with low BDE values, such as derivatives of acridine and heavy atom Se, which could not benefit the operational stability.

Most notably, as mentioned in the introduction of main text, some researchers hold the opposite view that degradation process in singlet state could be more vital (*Phys. Chem. Chem. Phys.* **21**, 438–447 (2019); *ACS Appl. Mater. Inter.* **12**, 31706–31715 (2020); *ACS Appl. Mater. Inter.* **14**, 22332–22340 (2022)), since unlike PH materials, the final emission of TADF emitters is from singlet state not triplet. **It seems that the degradation mechanism of TADF materials is still controversial and needs deeper insight. Thus, in this work, we systematically investigated the chemical degradation processes of typical blue TADF emitters at excited states, especially via**

the rigorous comparison experiments, we figured out that the cleavage of fragile bonds of TADF emitters at T_1 state is indeed the main cause of chemical degradation. More importantly, we uncover that $BDE-E_{T_1}$ as the critical molecule parameter of TADF emitters is positively and linearly correlated with the logarithm of quencher formation rate k_{QF} and this correlation also applies to corresponding device lifetime for a wide variety of TADF emitters. The new parameter, “ $BDE-E_{T_1}$ ”, emphasizes the importance of high BDE in stable TADF materials. Just as commented by reviewer #2, “*This work conducts fundamental and comprehensive studies on possible fundamental mechanisms of material factors potentially governing operation lifetimes of triplet-harvesting blue OLEDs and OLED emitters, particularly the advanced TADF blue emitters, and report an interesting discovery on the correlation between the difference between bond dissociation energy of fragile bonds and the first triplet state energy ($BDE-E_{T_1}$) and blue TADF emitter/OLED operation lifetimes.*”

Moreover, “ $BDE-E_{T_1}$ ” is not only a new and critical molecular parameter, but also easily accessible. BDE values could be easily obtained by calculating the ground state energy employing the widely used calculation methods. Compared with the commonly calculation method of investigating T_1 state stability, i.e., scanning potential energy surfaces of T_1 states with high precision method (*Adv. Opt. Mater.* **12**, 2102309 (2022)), calculating $BDE-E_{T_1}$ greatly lowers the calculation cost. And the nearly perfect linearity between $BDE-E_{T_1}$ and the activation energy of bond cleavage in T_1 state provide the solid theoretical foundation. **Importantly, the high reliability and low cost of $BDE-E_{T_1}$ for prediction the stability of TADF materials could be particularly valuable for high throughput screenings in material development with the aid of artificial intelligence.**

To address the Reviewer’s concern, we added more detailed discussions on Page 3 of main text as follows:

” ...Theoretical calculations on activation energy of the transition state during the bond cleavage solve the puzzle between thermodynamic parameter of $BDE-E_{T_1}$ and kinetic degradation process. This unique quantitative correlation actually applies to the device lifetime of a wide variety of blue TADF emitters reported by different research groups in exactly different device structures...”

“Considerable works have demonstrated that the operational stability of TADF-OLEDs can be enhanced by improving the material stability at singlet/triplet excited state. However, it is yet undetermined that whether the TADF emitters mainly degrade at singlet or triplet states, which still remains to be identified by rigorous and solid experiments based on a wide variety of TADF materials.”

Finally, we quite agree on the Reviewer’s statement that *find the solution of how to enhance the high $BDE-E_{T_1}$ in the T_1 states that’s more important for our current research (in OLEDs).*

Actually, we have been devoted to finding how to enhance BDE for a decade and achieved some valuable results. To promote our work, we revised the manuscript on Page 13 of the main text as follows: “...To enhance the “longevity gene” BDE- E_{T1} , one could start with two respects, i.e., lowering E_{T1} or improving BDE. For example, Adachi et al. introduced pyrene unit into MR-TADF material BCzBN, significantly lowering its E_{T1} and improving the operational stability of corresponding material and device (*Adv. Opt. Mater.* **8**, 2000102 (2020)). ... In the last decade, we focused on the BDE and intrinsic stability of blue organic emitters and drew some valuable design strategies for robust blue materials. We first confirmed the effective protection of fragile bonds in a ring is an efficient strategy to enhance BDE (*J. Phys. Chem. C* **118**, 7569–7578 (2014)) and next, conducted a systematic theoretical study on a series of carbazole(Cz)-based molecules with typical D- π -A structures and revealed that introducing electron withdrawing groups (EWGs) on 3,6, 2,7, or 4 positions of Cz would increase the BDE of the exocyclic C-N bond from Cz, while electron donating groups (EDGs) would decrease the BDE (*Chem. Mater.* **30**, 8771–8781 (2018)). Furthermore, Furthermore, to improve BDE in anion state (BDE(-)), we proposed a general strategy that introducing the delocalizing EWG (D-EWG) as negative charge manager within the molecule could significantly promote the BDE(-) by ~1 eV, and in particular, the introduction of D-EWG into the acceptor of a TADF molecular would also slightly lower the E_{T1} at the same time (*CCS Chem.* **4**, 331–343 (2022)). We believe that effective strategy to improve BDE without negative effects on material photophysical properties would be highly desired in the future development of robust blue TADF materials.”

Also, we have taken the critical comments seriously and have used the Reviewer’s constructive suggestions as a guide to further improve and strengthen our manuscript.

1. Authors mentioned the numerical simulation excluded the rate of SSA, STA, TTA because of lower rate compared to radiation, rISC, etc. However, there is no sufficient explanation of what changes occur in the simulation results, when the hot excitons are formed by TTA, TSA, SSA.

We thank the Reviewer for this question and are regrettable for the misunderstanding which might originate from our ambiguous description.

Firstly, on page 7 in the main text, we stated that “...according to simulation results, we excluded the effects of the reported quencher formation pathway, SSA, STA, and TTA process in our study.” We meant that we excluded **the effects of SSA, STA, and TTA on the quencher formation**. Actually, in our simulation, the SSA, STA, and TTA processes are considered, as in

equation (1)
$$\frac{dn_S}{dt} = I - (k_{r,S} + k_{nr,S} + k_{ISC})n_S + k_{RISC}n_T - (k_{SSA}n_S^2 + k_{STA}n_Sn_T -$$

$\gamma k_{\text{TTA}} n_{\text{T}}^2) - k_{\text{QS}} n_{\text{S}} n_{\text{Q}}$ and equation (2) $\frac{dn_{\text{T}}}{dt} = k_{\text{ISC}} n_{\text{S}} - (k_{\text{r,T}} + k_{\text{nr,T}} + k_{\text{RISC}}) n_{\text{T}} - (1 + \gamma) k_{\text{TTA}} n_{\text{T}}^2 - k_{\text{QT}} n_{\text{T}} n_{\text{Q}}$, we indeed incorporated the SSA, STA, and TTA terms. Just because at the experimental condition of illumination intensity $\sim 2 \text{ mW cm}^{-2}$ (Fig. 4b), the calculated rate of SSA, STA and TTA is 4-7 order of magnitude slower than that of radiation, RISC, etc., **the quenchers formed through SSA, STA, and TTA are negligible**. Accordingly, we write the equation (3) as $\frac{dn_{\text{Q}}}{dt} = k_{\text{QF}} n_{\text{T}}$, rather than $\frac{dn_{\text{Q}}}{dt} = k_{\text{QF}} n_{\text{T}}^2$, etc. Following the Reviewer's suggestion, we revised the statement on page 7 in the main text as "...according to simulation results, we excluded the effects of SSA, STA and TTA processes on the quencher formation in our study."

Secondly, on page S23 in supplementary information, we further discussed the comparison between single exciton model (quenchers mainly formed at T_1 state) and hot exciton models (quenchers mainly formed by SSA, STA, or TTA) in details as follows.

Take TTA as an example, in the hot exciton model (TTA), equation (3) of quencher formation changes to $\frac{dn_{\text{Q}}}{dt} = k_{\text{QF}}' n_{\text{T}}^2$. Compared with the equation (3) of $\frac{dn_{\text{Q}}}{dt} = k_{\text{QF}} n_{\text{T}}$, we could find that the rate of quencher formation is more sensitive to the change of triplet density (n_{T}^2 vs n_{T}), which would lead to the large degradation rate at the beginning part of degradation (\sim tens of minutes). And as the accumulation of quenchers, the degradation rate of hot exciton model would slow down quickly at the ending part (4~5 h) since the larger quencher formation rate. Thus, if we let the degradation rate in hot exciton model close to that in single exciton model at the beginning part, it would be smaller at the ending part, and vice versa, if we let the degradation rate in hot exciton model close to that in single exciton model at the ending part, it would be larger at the beginning part. Indeed, as shown in Fig. 4c, compared with the single exciton model, no matter how we adjusted the parameters in hot exciton model, the model could only fit well either at the beginning part (hot exciton model 1) or the ending part (hot exciton model 1) of the experimental results, which further supports that TTA process is not the main quencher formation way. When the hot excitons were formed via SSA or STA, the equation (3) would change to $\frac{dn_{\text{Q}}}{dt} = k_{\text{QF}} n_{\text{S}}^2$ or $\frac{dn_{\text{Q}}}{dt} = k_{\text{QF}} n_{\text{S}} n_{\text{T}}$. The degradation rate in hot exciton models (SSA or STA) would also much larger than that in single exciton model at the initial part and would also be more sensitive to the density of excitons. Thus, the similar degradation behaviors and simulation results of SSA and STA models with TTA model could be expected. Of note, we emphasized again that the neglect of SSA, STA, and TTA process is based on our experiment condition that the illumination intensity is $\sim 2 \text{ mW cm}^{-2}$ and the density of singlet and triplet excitons is $10^{13} \sim 10^{14} \text{ cm}^{-3}$, when the illumination intensity increases to $\sim 10^3 \text{ mW cm}^{-2}$ and the density of excitons increases to $10^{16} \sim 10^{17} \text{ cm}^{-3}$, the effect of SSA, STA, and TTA could no longer be ignored."

Fig. 4c comparison of different simulation results of single molecule and hot exciton models. with the assumption of $k_{nr,T}=0$.

2. When authors check the correlation between BDE- E_{T1} and device lifetime of 32 reported materials, the value of R^2 is just 0.44. Authors said it is because of device optimization and engineering or the specific molecule structures. Please explain why these materials have different results.

We thank the Reviewer for bringing up this point. We provided the related discussions of materials exhibited exceptionally longer lifetime beyond the correlation on page S15 in supplementary information. Here we would like to give further discussions and explanations as follows.

First and foremost, the operational stability of TADF devices is a complex issue. We are starkly aware of the fact that we could not employ one molecule parameter to fully describe the degradation of devices. Thus, we are really encouraged by the relatively well linear correlation between BDE- E_{T1} and the device lifetime of 22 materials reported by different research groups with $R^2=0.82$, which is never reported before, and we are not particularly disappointed by the unfortunate fact that R^2 decrease to 0.44 in 32 materials. As mentioned on page 11 of main text, for all of the materials, **the general trend remains the higher BDE- E_{T1} is, the longer device lifetime is.** And we provided supplementary discussions on page S15 in the supplementary information that for the materials exhibited exceptionally longer lifetime beyond the correlation, although their device lifetime is higher than predicted, materials with higher BDE- E_{T1} indeed showed longer device lifetime, such as DBA-DI vs TDBA-DI reported in *Adv. Opt. Mater.* **8**, 2000102 (2020); *Nat. Photon.* **13**, 540–546 (2019) (in the red circle), and TMCzTrz vs 5CzTrz reported in *Nat Photon.* **14**, 636–642 (2020) (in the blue circle). These results strongly revealed the degradation mechanism of TADF materials have general characteristic in essence and the thermodynamic parameter BDE- E_{T1} has the capability to be the shared longevity gene responsible for robust TADF materials and devices.

Supplementary Fig 16 | The correlation between $BDE_f - E_{T1}$ and device lifetime of 32 reported materials (35 points).

As for the reason the Review concerned. Our analysis and discussions are as follows.

Firstly, in principle, since RISC is a competition process for bond cleavage at T_1 state, the kinetic parameter k_{RISC} would also be important for improving molecule intrinsic stability or device lifetime. Indeed, Lee et al. demonstrated a good linearity correlation between k_{RISC} of TADF emitters and corresponding device operational lifetime (*Chem. Eng. J.* **427**, 130988 (2022)). (Fig. S18a). Yet, this correlation was obtained from 4 TADF emitters featuring very similar molecule structures. To explore its generality, we tried to collect k_{RISC} values of the above 35 points from literatures, but got only that of 13 points because the others were not reported. Unfortunately, we didn't find likewise good linearity correlation in such a wide variety of materials (Fig. S18c). It might originate from the high sensitivity of k_{RISC} value to different measurement and calculation methods, according to *J. Phys. Chem. A* **125**, 8074–8089 (2021). This result demonstrated again that **as for the “trait” of device lifetime, thermodynamic parameter $BDE - E_{T1}$ can act as the intrinsic molecule “longevity gene”, which is particularly valuable for high throughput virtual screening and material design. During “gene expression”, “environment” such as carrier balance or host-guest interactions, would also have important influence on the “trait” of device lifetime because they could greatly affect the kinetic molecule parameters such as k_{RISC} .** For DBA-DI, TDBA-DI, and 5CzTrz, they all possess high k_{RISC} values ($6.21 \times 10^6 \text{ s}^{-1}$, $1.08 \times 10^6 \text{ s}^{-1}$, and $1.5 \times 10^7 \text{ s}^{-1}$, respectively). Therefore, the longer operational lifetime of these 3 materials could also be partially attributed to the superior k_{RISC} .

Secondly, 9 points reported by Zhang et.al in *ACS Appl. Mater. Inter.* **12**, 31706–31715 (2020); *ACS Appl. Mater. Inter.* **14**, 22332–22340 (2022) (in the green circle) also did not fit the correlation well. These materials have very comparable BDE values ($\sim 4.2 \text{ eV}$ for carbazole derivatives and $\sim 3.6 \text{ eV}$ for DMAC derivatives) and E_{T1} values ($\sim 2.7 \text{ eV}$), which leads to very comparable BDE-

E_{T1} values (~ 1.5 eV for carbazole derivatives and ~ 0.9 eV for DMAC derivatives). Therefore, in the devices based on these materials, the difference in device lifetime is no longer decided by BDE- E_{T1} , but other factors. Indeed, Zhang et. al explained the different device lifetime by different k_f and EL emission wavelength of those carbazole derivatives in different host materials.

To improve the presentation and address the concerns raised by the Reviewer, we added a condensed version of the above discussions to page 11 in main text as “**Particularly, we carefully examined the materials exhibited exceptionally longer lifetime beyond the correlation. For DBA-DI and TDBA-DI reported by Kwon et. al^{20,49}, although their device lifetime is higher than predicted, DBA-DI with higher BDE- E_{T1} indeed showed longer device lifetime. Similar phenomenon occurred in TMCzTrz, 5CzTrz, and DACT-II reported by Adachi et.al³. And DBA-DI, TDBA-DI, and 5CzTrz, all possess high k_{RISC} values (6.21×10^6 s⁻¹, 1.08×10^6 s⁻¹, and 1.5×10^7 s⁻¹, respectively). Therefore, the longer operational lifetime of these 3 materials could also be partially attributed to the superior k_{RISC} and the prudent device optimization and engineering. On the other hand, we found 9 materials reported by Zhang et.al^{24,25} also did not fit the correlation well. These materials have very comparable BDE values (~ 4.2 eV for carbazole derivatives and ~ 3.6 eV for DMAC derivatives) and E_{T1} values (~ 2.7 eV), which leads to very comparable BDE- E_{T1} values (~ 1.5 eV for carbazole derivatives and ~ 0.9 eV for DMAC derivatives). Therefore, in these materials, the difference in device lifetime is no longer decided by BDE- E_{T1} , but other factors. Indeed, Zhang et. al explained the different device lifetime by different k_f and EL emission wavelength of those carbazole derivatives in different host materials²⁴ (See Discussions of materials exhibited exceptionally longer lifetime beyond the correlation in supplementary information for more detailed discussions.)**”

3. *Authors, please check the supporting page S21 line 279; I think Fig. 3c should be changed to Fig 4c.*

We thank the Reviewer for the careful reading. Following the Reviewer's suggestion, we have changed Fig. 3c to Fig. 4c on page S23 in Supplementary information as “...**Indeed, as shown in Fig. 4c, compared with the single exciton model...**”

We also appreciate to the Reviewer for all the critical comments that have allowed us to assemble an improved, stronger manuscript and hope that the Reviewer is satisfied with our revisions and detailed responses.

Reviewer #2 (Remarks to the Author):

Stable and long lifetime triplet-harvesting blue OLEDs and OLED emitters (phosphorescent, TADF etc.) have been a long sought but un-solved issue for both research and industry. This work conducts fundamental and comprehensive studies on possible fundamental mechanisms of material factors

potentially governing operation lifetimes of triplet-harvesting blue OLEDs and OLED emitters, particularly the advanced TADF blue emitters, and report an interesting discovery on the correlation between the difference between bond dissociation energy of fragile bonds and the first triplet state energy (BDE-ET1) and blue TADF emitter/OLED operation lifetimes. This reviewer finds this is a significant finding and contribution that would be highly useful for guiding development of highly stable blue TADF emitters and OLEDs. The results and evidence provided are considered convincing. I would recommend publication of this contribution.

We are very grateful to the Reviewer for the positive views on our manuscript and highlighting the key findings of our work.

As underlined by the Reviewer, *Stable and long lifetime triplet-harvesting blue OLEDs and OLED emitters (phosphorescent, TADF etc.) have been a long sought but un-solved issue for both research and industry.* In our work, via in-material chemistry, we comprehensively studied the degradation on the advanced TADF materials and demonstrated chemical degradation of TADF molecules involves the critical role of bond cleavage at triplet state rather than singlet and disclosed the difference of bond dissociation energy of fragile bonds and the first triplet state energy (BDE- E_{T1}) is linearly correlated with the logarithm of reported device lifetime for a wide variety of TADF emitters. We believe our work would pave a new avenue for high-throughput screenings and rational design of robust TADF emitters and to speed up the iteration and commercialization of highly efficient and stable OLED material and devices.

Reviewer #3 (Remarks to the Author):

The manuscript provided the correlation between theoretical calculated bond dissociation energy, exciton lifetime, photo-stability and OLED device stability. To improve the stability especially on the blue OLED, this type study attracts much interest from OLED researchers. However, there are several questions as below. To publish in the Nature communications, the major correction would be necessary.

1. Basically, the delayed emission quenching of TADF is not perfect in aerated condition. It is because that the triplet quenching is basically electron exchange between excited molecule and oxygen as explained in Dexter energy transfer, therefore, the concentration should be an important factor. The authors explained as “2-BP has higher S1 energy (3.45 eV) and lower T1 energy (2.47 eV) than these TADF emitters’ (2.7~3.0 eV), so it cannot quench those S1 excitons but efficiently quench T1 excitons. Indeed, the emission intensity of each degassed solution with 2-BP were nearly unchanged (Fig 2b), corroborating our assumption” in main text. This explanation may induce misunderstanding it related to be the energy state. The quencher concentration for solution state for Fig 2b should be provided. Also, the quencher should not diffuse in the film, which makes the difference between in solution and film states. Therefore, these result in solution and films should not be compared. In addition, the quencher is 2-BP in the main text and figure caption but NPK in figure.

First and foremost, we are very grateful to the Reviewer for the overall positive view on our study and we also appreciate the Reviewer for these helpful concerns that allow us to make better comparison in our revised manuscript.

1) Following the Reviewer's suggestion, we have clearly stated the quencher concentration for solution state in the figure captions of **Fig. 2b** as “The quencher concentration for these solutions is 5.0×10^{-4} M.” and we corrected the abbreviation of the quencher as 2-BP in Fig. 2c and 2d.

2) We totally agree with the Reviewer's comments that *these result in solution and films should not be compared*. And we revised the manuscript on page 5 in the main text as “**Notably, the degradation in films can't be completely suppressed even in TSF films or quencher-doped film because the emitter or quencher could not effectively diffuse in the film.**”

2. About Fig S3 and S4, authors compared the diluted ($\lambda_{380} = 0.167$) and concentrated ($\lambda_{380} = 0.993$) to check the influence of singlet exciton for photo-degradation. It is well known things that the photo degradation occurs from triplet exciton mainly because of its longer lifetime than singlet exciton. The reaction probability is related not only the concentration but also the excitation state lifetime, therefore, it should be measured with the higher S1 exciton density than that of T1 (during all the time for steady state excitation) with the consideration of those time dependency. While the resulted conclusion might not be different, is the measurement system satisfied this condition? In addition, reviewer have worried about its measurement system. When absorbance around 1 with $10 \times 10 \text{ mm}^2$ cuvette was employed, it provides the large gradation of exciton concentration in solution. This would affect to the photo-degradation result. The measurement should be investigated with the same absorbance with the difference light path length. For example, it is necessary to use the cell having the light path length as 10 mm for diluted condition and 1 mm for concentrated condition in the case of 10 times difference concentration.

We thank the Reviewer for these important suggestions.

Firstly, we quite agree that *the photo degradation occurs from triplet exciton mainly because of its longer lifetime than singlet exciton*. In the degassed solution without triplet quencher (no matter diluted or concentrated), the material degrades significantly because the T₁ excitons with $\sim \mu\text{s}$ lifetime exist, while in the aerated solution or solution with triplet quenchers, the long lifetime T₁ excitons are quenched, and thus, the PL intensity after photo-degradation is nearly unchanged. It is well known that the longer lifetime of T₁ exciton than S₁ exciton is originated from the much slower k_{RISC} (10^{4-6} s^{-1}) compared with k_{ISC} (10^{6-7} s^{-1}). Meanwhile, in the degassed solution without triplet quencher, an equilibrium would establish in the conversion between S₁ and T₁ exciton. Under this circumstance, the T₁ exciton density $[T_1] = [S_1] \frac{k_{\text{ISC}}}{k_{\text{RISC}}}$, leading to much larger T₁ than S₁ exciton density [S₁]. **Thus, one could find that the longer lifetime and larger exciton density of T₁ exciton compared with S₁ exciton are two manifestations of the same thing of $k_{\text{RISC}} \ll k_{\text{ISC}}$ from different aspects.** When things come to aerated solution or solution with high concentration quencher, the equilibrium between S₁ and T₁ no longer exists because of the much larger quenching rate constant, which could be verified by the vanishment of delayed fluorescence in the transient PL spectra. Under this circumstance, T₁ excitons are almost all quenched and so the concentration of

T_1 exciton [T_1] is much smaller than S_1 exciton concentration [S_1]. Thus, the reviewer suggested that *it should be measured with the higher S_1 exciton density than that of T_1 (during all the time for steady state excitation) with the consideration of those time dependency*. Actually, the experiment results of aerated solution or solution with triplet quenchers could exactly be an appropriate reflection of the condition that S_1 exciton density is higher than that of T_1 during all the time for steady state excitation.

Secondly, we fully understand the Reviewer's concern of *the large gradation of exciton concentration in solution would affect to the photo-degradation result* and sincerely thank the Reviewer for bring up the point. Following the Reviewer's suggestion, we customized the cuvettes with light path length of 1 mm, and conducted photo-degradation tests of concentrated solution in 1 mm cuvette and diluted solution in 10 mm cuvette with the same absorption about 0.104. We added this experiment results in Supplementary Fig. 4 and Supplementary Discussions for photo-degradation tests in solution on Page S6 in Supplementary information as follows.

“Moreover, in the concentrated solution, the large gradation of exciton concentration might exist since the cuvette is $10 \times 10 \text{ mm}^2$. To alleviate this undesirable effect, we further customized cuvettes with light path length of 1 mm (Supplementary Fig. 4a), and conducted the photo degradation tests of DMAC-DPS with concentrated solution in 1 mm cuvette and diluted solution in 10 mm cuvette with the same absorption of 0.104 (Supplementary Fig. 4b). As shown in Supplementary Fig. 4c, the initial PL intensity of aerated and concentrated solution with 1 mm light path is $\sim 1/2$ of that of diluted solution (228 and 415). Considering the fact that concentrated solution only has $1/10$ emitting region of diluted solution. This PL intensity difference indicated that the density of S_1 exciton in the aerated and concentrated solution would be larger than that of in degassed and diluted solution. Meanwhile, in the $1 \times 10 \text{ mm}^2$ cuvette, the gradation of exciton concentration would be alleviated. Thus, the degradation results in Supplementary Fig. 4d that the PL intensity of concentrated solution almost did not change since the T_1 excitons were quenched while the PL intensity of degassed dilute solution shows visible decay, strongly confirmed that TADF materials mainly degrade at T_1 state once again.”

Supplementary Fig. 4 | Degradation tests of dilute and concentrated solution in different light path length. a Cuvettes with light path of 1 mm (left) and 10 mm (right). b, c, d Absorption spectra (b), PL spectra (c), and UV degradation results under 380 nm illumination (d) of concentrated solution (5.0×10^{-4} M) in 1×10 mm² cuvette and diluted solution (5.0×10^{-5} M) in 10×10 mm² cuvette.

3. The ISC and RISC rate constants are affected by how PF/DF efficiencies are estimated, and whether non-radiative decay from S1 or T1 is assumed to be 0. Authors assuming the singlet non-radiative decay as 0 but it is reasonable to estimate with triplet non-radiative decay as 0 when radiative decay rate was assumed as 0. For more detailed discussion, it would be better to discuss with the range of rate constants. The detail of photophysical analysis for TADF was reported in *J. Phys. Chem. A* 2021, 125, 8074-8089. Especially, the materials showing 100% internal quantum efficiency in OLED have no non-radiative decay from triplet as reported in *Front. Chem.* 2022, 10, 990918.

Thanks the Reviewer for the valuable suggestion. According to the photophysical analysis for TADF reported in *J. Phys. Chem. A* **125**, 8074-8089 (2021), we further conducted the simulation with the assumption of $k_{nr,T} = 0$, and got a bigger k_{QF} value of each material. The simulation results were summarized in Supplementary Table 1 and Supplementary Fig. 7. Together with the smaller k_{QF} value with the assumption of $k_{nr,S} = 0$, we got the range of the k_{QF} value. Fig. 4d shows the plot of average value of k_{QF} as a function of BDE- E_{T1} . To address the review's concern, we have revised the manuscript on page 7 of the main text as follows:

“Since the calculated photophysical parameters depend on whether non-radiative decay from

S_1 or T_1 is assumed to be 0 (*J. Phys. Chem. A* **125**, 8074-8089 (2021)), we got two k_{QF} values of each material with the assumption of $k_{nr,T} = 0$ and $k_{nr,S} = 0$, respectively.”

“...and found the logarithm of k_{QF} is negatively and linearly correlated with the value of $BDE-E_{T1}$ whether $k_{nr,T}$ (Supplementary Fig. 7c) and $k_{nr,S}$ (Supplementary Fig. 7d) is assumed to be 0. And the logarithm of average value of k_{QF} is correlated with $BDE-E_{T1}$ with $R^2=0.88$ (Fig. 4d), ...”

Supplementary Fig 7 | **a** Experimental and simulation results on neat films of DMAC-DPS, SpiroAC-Trz, DPAC-Trz, DCzTrz, DDCzTrz, and 5CzBN with the thickness of 80 nm. **b** Comparison of experimental (dots) and simulation (line) results of τ_p change in photo-degradation tests. **c** The correlation between the quencher formation rate (k_{QF}) and $BDE-E_{T1}$. All results were simulated with the assumption of $k_{nr,S}=0$. **d**, The correlation between the quencher formation rate (k_{QF}) and $BDE-E_{T1}$ (c) with the assumption of $k_{nr,T}=0$.

Fig. 4d | The correlation between the average quencher formation rate (k_{QF}) and $BDE-E_{T1}$.

4. About the hot-exciton model: The Fig 3 was illustrated as all of SSA, STA, TTA provide the S₁ excitons however only TTA provided the S₁ exciton in equation 1. Also, it would not suitable TTA efficiency was assumed as 0.25 because the TT up-conversion efficiency related to be each molecule. How does the simulation result obtain when TT up-conversion does not happen? Also, there are no term for $k(QF)n(T)$ in equation 2.

We are grateful to the Reviewer for these constructive suggestions that help us shed more light on the effect of hot excitons.

1) We thank the Reviewer for pointing the inaccuracy in Fig. 3 and are regrettable for this ambiguous description. In the revised manuscript, we have renewed Fig. 3 with more accurate description of SSA/STA/TTA process as follows. Usually, SSA process consumes two S₁ excitons and produce one S₁ exciton, thus the coefficient of SSA process in equation (1) is 2-1=1 and STA process consumes one S₁ exciton, one T₁ exciton and produce one T₁ exciton, thus the coefficient of STA process in equation (2) is 0 while in equation (1) is 1 according to *PRL*, **108**, 267404 (2012).

Fig 3 | Jablonski diagram of the exciton dynamics in TADF materials by photo-excitation.

2) As for TTA process, it would consume two triplet exciton and produce γ S₁ exciton and (1- γ) T₁ exciton. For all the TADF materials studied in our work with $E_{S1} < 2E_{T1}$, and $E_{T2} < 2E_{T1}$, γ is 0.25 accordingly to the literatures such as *Org. Electron.*, **14**, 2721–2726 (2013), *Adv. Funct.*

$$T_1 + T_1 \rightarrow \frac{1}{4}S + \frac{3}{4}T.$$

3) To address the Review’s concern, we added corresponding simulation results in Supplementary Fig. 8 and statement on page 7 in the main text as follows “we conducted the simulation with/without TTA process under the assumption of $k_{nr, T} = 0$ and assumption of $k_{nr, S} = 0$ (Supplementary Fig. 8), respectively. Regardless of $k_{nr, T} = 0$ or $k_{nr, S} = 0$, the simulation results with/without TTA process are nearly unchanged, which indicates TTA process have little effect on the degradation of materials in our work.”

Supplementary Fig. 8 | Simulation results of DMAC-DPS, SpiroAC-Trz, DPAC-Trz, DCzTrz, DDCzTrz, and 5CzBN with/without TTA process under the assumption of $k_{nr, T} = 0$ (a), and $k_{nr, S} = 0$ (b).

4) We thank the Reviewer for the good point that *there is no term for $k(QF)n(T)$ in equation 2.*

Indeed, the quencher formation is also a consumption way of T₁ excitons. In the revised manuscript, we have added $k_{\text{QF}}n_{\text{T}}$ in equation (2) as follows:

$$\frac{dn_{\text{T}}}{dt} = k_{\text{ISC}}n_{\text{S}} - (k_{\text{r,T}} + k_{\text{nr,T}} + k_{\text{RISC}})n_{\text{T}} - (1 + \gamma)k_{\text{TTA}}n_{\text{T}}^2 - k_{\text{QT}}n_{\text{T}}n_{\text{Q}} - k_{\text{QF}}n_{\text{T}} \quad (2)$$

Notably, we have renewed the simulation results with $k_{\text{QF}}n_{\text{T}}$ in equation (2), and the simulation results didn't change, because k_{QF} is $\sim 10^{-3}$, 7-9 order of magnitude smaller than $k_{\text{nr,T}}$, k_{RISC} , and $k_{\text{QT}}n_{\text{Q}}$ ($\sim 10^{4-6}$). Thus, the T₁ exciton consumption way of quencher formation could be neglected. This results further confirmed that the decay of PL is originated from quenching by quenchers other than direct vanishing of emitters and the very few quenchers formed by irreversible bond cleavage indeed could have deadly influence on degradation of material and devices.

5. The experimental photostability was investigated in neat film but the material was doped in host-matrix in the device. Also, the calculation is the mono molecular state. Is it possible to compare them directly?

First of all, we thank the Reviewer for encouraging us to pursue more possibility.

Indeed, in the devices, the TADF materials are doped in host materials with wide band gap and high E_{T1} , such as mCBP. However, mCBP also possess fragile C-N bonds and high T₁ energy, which might lead to undesired effects in the degradation tests. Thus, in this study, to separate variables and investigate the intrinsic stability of TADF materials themselves, we choose neat film to conduct the photo-degradation test and simulation at first via in-material chemistry. Encouragingly, we got the critical molecular parameter BDE- E_{T1} for the intrinsic stability of TADF materials. And the marvelous correlation between BDE- E_{T1} and device lifetime strongly revealed that the degradation mechanism at T₁ state have the general characteristic in essence and the BDE- E_{T1} has the ability to be the shared longevity gene responsible for robust TADF materials and devices again.

Also, in this study, the calculation is performed for mono molecule, which is limited by the relatively low accuracy of calculation for aggregation state while the relatively high accuracy requirement for BDE calculation. We have the great desire that the development of molecular dynamics to help us get more insight into the aggregation state of TADF materials in the future. Importantly, the BDE calculation for mono molecular indeed could reflect the relative strength of chemical bonds. In our previous work, we have gotten BDE values calculated in mono molecular state for typical OLED materials and the calculated results were verified by the MALDI-TOF-MS for neat films (*J. Phys. Chem. C*, **116**, 19451-19457 (2012); *J. Phys. Chem. C*, **118**, 7569-7578 (2014); *CCS Chem.* **4**, 331-343 (2022)).

Most importantly, the calculation in mono molecule has the superiority in computational cost and provide high feasibility for HTVS in material development to speed up the iteration and commercialization of highly efficient and stable blue OLED materials and devices.

6. *There is the mistake in the reference number. Also, the style of references is not unified. It should be check again carefully.*

We thank the Reviewer for the careful reading. We have carefully checked the reference number and style (Author list. Title of paper in sentence case. *Name of journal* volume number, initial-final page numbers or article number (year).) in the revised manuscript.

We appreciate the Reviewer for the overall positive remarks, and the valuable comments and constructive suggestions that have helped us to assemble a stronger manuscript and sincerely hope that the reviewer is satisfied with our detailed responses.

REVIEWERS' COMMENTS

Reviewer #1 (Remarks to the Author):

Revised manuscript is well revised as per reviewer comments.

I recommend publishing as it is.

Reviewer #3 (Remarks to the Author):

The manuscript would be almost suitable to publish in Nature Communications except the several points commented in below.

Figure S4:

There is no information which material was used to this experiment.

Rate equations:

Authors provided the several equations to estimate rate equations referring with J. Phys. Chem. A 125, 8074 (2021). However, the rate equations for TADF are eventually explained as like in attached word file as a universal in that paper. For the delayed emission efficiency authors explain with sum of $(\Phi_{ISC}\Phi_{RISC})^i$ from $i=0$ to ∞ , but it should be from $i=1$ to ∞ . Because $\Phi_p(\Phi_{ISC}\Phi_{RISC})^i = \Phi_p$ when $i=0$, original equation authors provided means Φ_{PLQY} .

(See attached word file because of equations containing.)

Table S4:

The writing should be $A \times 10^B$ as the similar format with others.

Style in literatures:

The style of author of literature is fluctuating; some of literature provided the full authors but some of literature provides the abbreviated authors with et al.

Rate equations:

Authors provided the several equations to estimate rate equations referring with J. Phys. Chem. A 125, 8074 (2021). However, the rate equations for TADF are eventually explained as like below as a universal in that paper.

$k_p = \tau_p^{-1} = k_r + k_{nr,S} + k_{ISC} \left(1 + \frac{k_{RISC}}{k^S - k_d} \right)$	
$k_d = \tau_d^{-1} = k_{r,T} + k_{nr,T} + k_{RISC} \left(1 - \frac{k_{ISC}}{k^S - k_d} \right)$	
$\Phi_p = \frac{k_r}{k_p} = \frac{(A_p + A_d)k_d}{A_p k_d + A_d k_p} \Phi_{PLQY}$	
$\Phi_d = \Phi_p \sum_{i=1}^{\infty} (\Phi_{ISC} \Phi_{RISC})^i = \frac{A_d(k_p - k_d)}{A_p k_d + A_d k_p} \Phi_{PLQY}$	
With the limit condition of $k_{nr,T} = 0, k_{r,T} = 0$	With the limit condition of $k_{nr,S} = 0, k_{r,T} = 0$
$k_{ISC} = k_p \frac{\Phi_d}{\Phi_{PLQY}} + k_d \frac{\Phi_d}{\Phi_p}$	$k_{ISC} = k_p (1 - \Phi_p)$
$k_{RISC} = k_d \frac{\Phi_{PLQY}}{\Phi_p}$	$k_{RISC} = \frac{k_p k_d \Phi_d}{k_{ISC} \Phi_p}$

In here, A_p and A_d are the pre-exponential factor of fitting curve for prompt and delayed components, and k^S is the singlet exciton decay rate explaining as $k_p - k_d(\Phi_d/\Phi_p) + k_{ISC}(\Phi_{r,T}/\Phi_{r,S})$. For the delayed emission efficiency authors explain with sum of $(\Phi_{ISC}\Phi_{RISC})^i$ from $i = 0$ to ∞ , but it should be from $i = 1$ to ∞ . Because $\Phi_p(\Phi_{ISC}\Phi_{RISC})^i = \Phi_p$ when $i = 0$, original equation authors provided means Φ_{PLQY} .

Point-by-point Response to the Reviewers' comments
Manuscript ID: NCOMMS-23-14158-A

We sincerely thank the Referees for their insightful comments, constructive suggestions, and the time spent in reviewing our manuscript. Please find below our point-by-point responses to the comments. To ease readability, we are repeating the referee's comments in *italics*.

Reviewers' Comments:

Reviewer #1 (Remarks to the Author):

Revised manuscript is well revised as per reviewer comments.

I recommend publishing as it is.

Response:

We are very grateful to the Reviewer for the recognition of our responses and recommendation of the publishing of our work.

Reviewer #3 (Remarks to the Author):

The manuscript would be almost suitable to publish in Nature Communications except the several points commented in below.

Figure S4:

There is no information which material was used to this experiment.

Response:

We thank the Reviewer for the careful reading. Following the Reviewer's suggestion, we have added the information of the material used in the photo-degradation in Supplementary Fig. 4 on page S5 as follows.

Supplementary Fig. 4 | Degradation tests of dilute and concentrated solution of DMAC-DPS in different light path length.

Rate equations:

Authors provided the several equations to estimate rate equations referring with *J. Phys. Chem. A* 125, 8074 (2021). However, the rate equations for TADF are eventually explained as like in attached word file as a universal in that paper. For the delayed emission efficiency authors explain with sum of $(\Phi_{ISC}\Phi_{RISC})^i$ from $i=0$ to ∞ , but it should be from $i=1$ to ∞ . Because $\Phi_p(\Phi_{ISC}\Phi_{RISC})^i = \Phi_p$ when $i=0$, original equation authors provided means Φ_{PLQY} . (See attached word file because of equations containing.)

Response:

We thank the Reviewer's for the careful reading. In fact, the equations in the Supplementary Information are derived from equations 52-57 in (*J. Phys. Chem. A* 125, 8074 (2021)) as follows:

The corresponding rate constants are thus described by eqs 52–57.

$$k_r^S = k_p \Phi_{PF} = k_p \Phi_r^S \quad (52)$$

$$k_{nr}^S = k_p \Phi_{nr}^S = k_p (1 - \Phi_r^S - \Phi_{ISC}) \quad (53)$$

$$k_{ISC} = k_p \Phi_{ISC} \quad (54)$$

$$k_r^T = k_d \Phi_r^{TOE} = k_d \frac{\Phi_{phos}}{\Phi_{ISC}} = k_d \frac{\Phi_{DE} (1 - R_{DE}^{DF})}{\Phi_{ISC}} \quad (55)$$

$$k_{nr}^T = k_d - (1 - \Phi_{ISC}) k_{RISC} - k_r^T \quad (56)$$

$$k_{RISC} = k_d \frac{\Phi_{RISC}^{OE}}{1 - \Phi_{ISC}} = \frac{k_p k_d}{k_{ISC}} \frac{\Phi_{DF}}{\Phi_r^S} = k_d \frac{\Phi_{DE} R_{DE}^{DF}}{\Phi_r^S \Phi_{ISC}} \quad (57)$$

During the derivation, we employed the assumption that $k_{r,S} + k_{nr,S} + k_{ISC} \gg k_{RISC}$ to approximate $k_p \sim k_s$. This assumption would influence the expressions of k_{ISC} and k_{RISC} with the assumption of $k_{nr,T} = 0$. We are very grateful to the Reviewer for pointing out the exact equations without the assumption that $k_{r,S} + k_{nr,S} + k_{ISC} \gg k_{RISC}$. Following the Reviewer's suggestion, we

renewed the equations on Page S9 with adding the detailed expression of Φ_p , Φ_d , k^S and revised the expressions of k_{ISC} and k_{RISC} with the assumption of $k_{nr,T} = 0$ as follows.

$I = \frac{P\lambda\cos\theta(1-T)}{hcd}$	
$\Phi_p = \frac{k_r}{k_p} = \frac{(A_p + A_d)k_d}{A_p k_d + A_d k_p} \Phi_{PLQY}$	
$\Phi_d = \Phi_p \sum_{i=1}^{\infty} (\Phi_{ISC} \Phi_{RISC})^i = \frac{(k_p - k_d)A_d}{A_p k_d + A_d k_p} \Phi_{PLQY}$	
$k^S = k_p - k_d \frac{\Phi_d}{\Phi_p} + k_{ISC} \frac{\Phi_{r,T}}{\Phi_{r,S}}$	
With the assumption of $k_{nr,T} = 0, k_{r,T}=0$	With the assumption of $k_{nr,S} = 0, k_{r,T}=0$
$k_p = \tau_p^{-1} = k_r + k_{ISC}(1 + \frac{k_{RISC}}{k^S - k_d}) + k_{nr,S}$	$k_p = \tau_p^{-1} = k_r + k_{ISC}(1 - \frac{k_{ISC}}{k^S - k_d})$
$k_d = \tau_d^{-1} = k_{RISC}(1 - \frac{k_{ISC}}{k^S - k_d})$	$k_d = \tau_d^{-1} = k_{nr,T} + k_{RISC}(1 - \frac{k_{ISC}}{k^S - k_d})$
$k_{ISC} = k_p \frac{\Phi_d}{\Phi_{PLQY}} + k_d \frac{\Phi_d}{\Phi_p}$	$k_{ISC} = k_p(1 - \Phi_p)$
$k_{RISC} = k_d \frac{\Phi_{PLQY}}{\Phi_p}$	$k_{RISC} = \frac{k_p k_d \Phi_d}{k_{ISC} \Phi_p}$

By employing the new expressions of k_{ISC} and k_{RISC} , we got new calculated results of them as follows (values in the parentheses are the previous results).

Material	DMAC-DPS	SpiroAC-Trz	DPAC-Trz	DCzTrz	DDCzTrz	5CzBN
τ_p (ns)	23.5	18.8	17.9	9.7	14.0	16.3
τ_d (μ s)	4.4	10.4	8.9	6.6	9.2	3.6
k_{ISC} (10^7 s^{-1})	3.57(3.47)	2.64(2.65)	2.57(2.58)	2.48(2.48)	2.84(2.85)	4.36(4.29)
$k_{nr,S}$ (10^6 s^{-1})	0.85(1.85)	3.49(3.40)	8.56(8.47)	67.1(67.1)	33.5(33.4)	8.12(8.76)
k_{RISC} (10^5 s^{-1})	12.2(12.2)	1.91(1.91)	2.08(2.08)	2.00(2.00)	1.79(1.79)	9.16(9.16)
$k_{nr,T}$ (10^4 s^{-1})	0	0	0	0	0	0
k_{QF} (10^{-3} s^{-1})	9.7 (10)	3.8 (3.8)	6.0 (6.0)	2.3 (2.3)	0.66(0.66)	0.40 (0.55)

With the new k_{ISC} , k_{RISC} , and $k_{nr,S}$ values, we reconducted the simulation with the assumption of $k_{nr,T}=0$, and got the new k_{QF} values. We found slight change in k_{ISC} , $k_{nr,S}$, and k_{QF} values for DMAC-DPS and 5CzBN because they both possess relatively larger k_{RISC} values. In this

circumstance, the assumption that $k_{r,S} + k_{nr,S} + k_{ISC} \gg k_{RISC}$ might not work well, while for SpiroAC-Trz, DPAC-Trz, DCzTrz, and DDCzTrz with relatively smaller k_{RISC} values, the k_{ISC} , $k_{nr,S}$, and k_{QF} values are nearly unchanged. We also renewed Supplementary Table 1 on Page S17 in Supplementary Information with these new values. According to the new k_{QF} values of DMAC-DPS and 5CzBN, we revised Fig. 4d on Page 8 in the main text as follows. We also renewed the corresponding simulation results with the assumption of $k_{nr,T}=0$ in Supplementary Fig. 7. Of note, though with the new k_{QF} values, R^2 of the correlation between $BDE-E_{T1}$ and $\text{Ln}(k_{QF})$ is slightly decrease from 0.88 to 0.83, results demonstrated in Fig. 4d could still strongly support the conclusion that “ $BDE-E_{T1}$ is really one key parameter determining the molecular stability of blue TADF materials at triplet states.”

Fig. 4d The correlation between the average quencher formation rate (k_{QF}) and $BDE-E_{T1}$. The error bar values refer to the two different k_{QF} values with the assumption of $k_{nr,T}=0$ (the upper bar) or $k_{nr,S}=0$ (the below bar).

Table S4:

The writing should be $A \times 10^B$ as the similar format with others.

Response:

We thank the Reviewer for bringing this point. Following the Reviewer's suggestion, we have revised the writing in Supplementary Table 4 as $A \times 10^B$ as follows:

Supplementary Table 4. The output simulation results of DMAC-DPS neat film

Time (s)	$n_Q(t)$ (cm^{-3})	$n_S(t)$ (cm^{-3})	$n_T(t)$ (cm^{-3})
...
8.51 × 10 ²	1.55 × 10 ¹⁵	5.79 × 10 ¹²	1.73 × 10 ¹⁴
1.03 × 10 ³	1.85 × 10 ¹⁵	5.61 × 10 ¹²	1.67 × 10 ¹⁴
1.33 × 10 ³	2.32 × 10 ¹⁵	5.35 × 10 ¹²	1.59 × 10 ¹⁴
1.62 × 10 ³	2.76 × 10 ¹⁵	5.12 × 10 ¹²	1.52 × 10 ¹⁴
...

Style in literatures:

The style of author of literature is fluctuating; some of literature provided the full authors but some of literature provides the abbreviated authors with et al.

Response:

We thank the Reviewer for the careful reading. According to the Nature style for References, when the number of authors is ≤ 5 , the author list would list the full authors, while when it is > 5 , the author list would be as “First author, et al.”, such as the recent work published in *Nature communications*. (*Nature Communications* 14, 419 (2023); *Nature Communications* 14, 2500 (2023)).

We appreciate the Reviewer for the valuable comments and constructive suggestions that have helped us to assemble a stronger manuscript and sincerely hope that the reviewer is satisfied with our detailed responses.